# Oracle-Efficient Online Learning for Smoothed Adversaries

Nika Haghtalab, Yanjun Han, Abhishek Shetty, and Kunhe Yang

University of California, Berkeley
{nika,yjhan,shetty,kunheyang}@berkeley.edu

## Abstract

We study the design of computationally efficient online learning algorithms under smoothed analysis. In this setting, at every step an adversary generates a sample from an adaptively chosen distribution whose density is upper bounded by $1/\sigma$ times the uniform density. Given access to an offline optimization (ERM) oracle, we give the first computationally efficient online algorithms whose sublinear regret depends only on the pseudo/VC dimension $d$ of the class and the smoothness parameter $\sigma$. In particular, we achieve *oracle-efficient* regret bounds of $O(\sqrt{Td\sigma^{-1}})$ for learning real-valued functions and $O(\sqrt{Td\sigma^{-\frac{1}{2}}})$ for learning binary-valued functions. Our results establish that online learning is computationally as easy as offline learning, under the smoothed analysis framework. This contrasts the computational separation between online learning with worst-case adversaries and offline learning established by [HK16].

Our algorithms also achieve improved bounds for some settings with binary-valued functions and worst-case adversaries. These include an oracle-efficient algorithm with $O(\sqrt{T(d|\mathcal{X}|)^{1/2}})$ regret that refines the earlier $O(\sqrt{T|\mathcal{X}|})$ bound of [DS16] for finite domains, and an oracle-efficient algorithm with $O(T^{3/4}d^{1/2})$ regret for the transductive setting.

## 1 Introduction

Adversarial online learning is a cornerstone of modern machine learning and has led to significant advances in computer science broadly. A recent line of work on "beyond the worst-case analysis" of online learning has brought into light the overly pessimistic nature of standard characterizations of online learnability [RST11, GR17, HRS20, HRS22]. This is exemplified by the results of [HRS22] showing that adversarial online learnability is *statistically* as easy as PAC learnability, in presence of noise. That is, under *smoothed analysis*, online and offline learnability are both characterized by the finiteness of the VC dimension of a hypothesis class as opposed to the much larger Littlestone dimension that characterizes online learnability in the worst-case [BDPSS09]. However, to fully deliver on the promise revealed by these statistical insights, there needs to be an algorithmic framework for realizing this connection between online and offline learnability. In this paper, we ask

> *whether efficient offline learning algorithms lead to efficient online learning algorithms with comparable regret guarantees, under smoothed analysis?*

In more detail, smoothed analysis is a perspective on algorithm design, introduced by [ST04] and formalized for online learning by [RST11, HRS20], in which the adversary is restricted to generating an instance at every round from a distribution that is not overly concentrated, i.e., a distribution whose

36th Conference on Neural Information Processing Systems (NeurIPS 2022).

| | | Regret Bound | Method | Reference |
|---|---|---|---|---|
| Statistical Upper Bound | Binary | $\widetilde{O}\left(\sqrt{dT\log(\sigma^{-1})}\right)$ | Prob. Coupling and $\epsilon$-Net | [HRS22, Thm 3.1] |
| | Real-values | $\widetilde{O}\left(\sqrt{dT\log(\sigma^{-1})}\right)$ | | Thm F.1 |
| Computational Upper Bound | Binary | $\widetilde{O}\left(\sqrt{dT\sigma^{-1/2}}\right)$ | FTPL with Poissonization 1 oracle call per round | Thm 3.2 |
| | Real/Binary-valued | $\widetilde{O}\left(\sqrt{dT\sigma^{-1}}\right)$ | Relax-and-Randomize 2 oracle calls per round | Thm 3.1 |
| Lower Bound | Alg-independent | $\Omega\left(\sqrt{T(d/\sigma)^{1/2}}\right)$ | Construction for runtime $o(\sqrt{d/\sigma})$ | Thm 5.1 |
| | Algorithms 1 and 2 | $\Omega\left(\sqrt{dT\sigma^{-1/2}}\right)$ | | Thm E.1 |
| Classical Settings | Small-domain | $O\left(\sqrt{T(d|\mathcal{X}|)^{1/2}}\right)$ | FTPL with Poissonization | Cor 3.3 |
| | Transductive learning | $O\left(T^{3/4}d^{1/4}\right)$ | | Cor 3.3 |

Table 1: In the above table, $d$ represents the pseudo dimension or VC dimension of the hypothesis class $\mathcal{H}$, $\sigma$ is the smoothness parameter, and $T$ is the number of time steps.

density is upper bounded by $1/\sigma$ times that of the uniform distribution[1]. The smoothness of the adversary's actions captures the noise and imprecision inherent in the real world. As $1/\sigma$ decreases, so does the uncertainty of the learner about the distribution of future instances. This gracefully captures the expressivity of worst-case instances while circumventing the overly pessimistic nature of the worst-case analysis.

The question of whether offline learning algorithms can lead to online learning algorithms is naturally captured by the *oracle-efficiency* framework (e.g., [DHL+20, KV05, HK16]). In this setting, we have access to an offline learning algorithm or equivalently an *empirical risk minimization (ERM)* oracle which can compute an optimal hypothesis given any history of the actions of the adversary, using $O(1)$ computation. Efficient algorithms must then be designed to tap into the existing ERM oracle, using polynomial number of calls and computation.

We aim to design oracle-efficient online algorithms whose regret resemble the statistically optimal regret as much as possible. In particular, under smoothed analysis, these bounds must be characterized by offline statistical complexity measures, such as the VC dimension or pseudo dimension of a hypothesis class. Interestingly, [HK16] showed that such computationally algorithms cannot exist for fully worst-case adversaries. Therefore, designing oracle-efficient online algorithms for adversaries who are not fully worst-case, must *simultaneously overcome both statistical and computational impossibilities.* Designing algorithms that achieve this is the main contribution of this paper.

## 1.1 Main Results

We consider online learning under smoothed analysis and give the first oracle-efficient online learning algorithms whose regret is characterized by the statistical offline complexity measures. In particular, we show that there are efficient algorithms, given access to an ERM oracle, that achieve sublinear regret that depends only on the pseudo- (or VC) dimension of a class of hypotheses, as well as a parameter that captures the power of the adversary, i.e., $\sigma$. We study both the real-valued and binary valued losses and achieve nearly tight upper and lower bound for these settings. We summarize our main results in Table 1.

**Upper bounds and Algorithms.** For the general real-valued case, we design an algorithm based on the *Relax-and-Randomize* principle of [RSS12] that achieves a regret bound of $O(\sqrt{Td\sigma^{-1}})$ for smoothed online learning, where $d$ is the pseudo-dimension of the hypothesis class. This algorithm uses 2 oracle calls per round. We improve these regret bounds for the binary classification setting under smoothed analysis and achieve regret of $O(\sqrt{Td\sigma^{-1/2}})$. The algorithm that achieves these improved bounds is a variant of FTPL that uses Poisson random variables in the design of its perturbations. This algorithm uses 1 oracle call per round. The improved regret in the binary setting crucially leverages the fact that smoothness over the instance domain $\mathcal{X}$ also implies smoothness over the instance-label pairs $\mathcal{X} \times \mathcal{Y}$, when the label set is binary (or constant in size).

---

[1]While we use the uniform distribution as the base measure for ease of exposition, our results also generalize to arbitrary known base measures.

While these bounds demonstrate sublinear regret that only depends on $d$ and $\sigma$, their dependence on $\sigma$ does not match the (non-efficient) statistically optimal regret bound of $\widetilde{O}(\sqrt{Td\ln(1/\sigma)})$. For the case of binary classification under smoothed analysis, the statistically optimal regret bound is due to [HRS22]. For the other settings, we provide the optimal statistical rates in Appendix F.

**Lower bounds.** We further investigate the gap between the computational and statistical regret upper bounds. We present an algorithm-independent lower bound for the setting of smoothed online learning that shows that any algorithm with runtime $o(\sqrt{d/\sigma})$ will incur a $\Omega(\sqrt{T(d/\sigma)^{1/2}})$ regret. We note that this lower bound demonstrates the same dependence on $1/\sigma$ as our upper bound (for the binary classification setting under smoothed analysis). We also provide algorithm-dependent regret lower bounds that demonstrate improved dependence on parameter $d$, and apply to all relax-and-randomize and FTPL-style algorithms.

**Improved Bounds for the Classical Settings.** In addition to constrained adversaries, our algorithms and analysis also imply improved regret bounds for two classical settings, namely, *online learning in bounded domains with worst-case adversaries* and traditional *online transductive learning*. For worst-case adversaries in binary classification with domain size $|\mathcal{X}|$, we achieve a regret of $O(\sqrt{T(d|\mathcal{X}|)^{1/2}})$, improving upon the $O(\sqrt{T|\mathcal{X}|})$ bound of [DS16]. For online transductive binary classification we achieve a regret bound of $O(T^{3/4}d^{1/4})$, improving upon the $O(T^{3/4}d^{1/2})$ bound in [KK05]. Both improvements are enabled by our novel Poissonized FTPL analysis and techniques which achieve stronger regret bounds in the binary classification setting.

## 1.2 Technical Overview

**Random Playout for Beyond Worst-Case Adversaries.** Our algorithms are based on the random playout design principles, including the *admissible relaxation* framework of [RSS12] and the *Follow-the-Perturbed-Leader* framework of [KV05]. We show that this framework is useful for analyzing online learning algorithms in the beyond worst-case setting, especially in smoothed analysis. In this setting, smoothness captures a level of predictability about the future. This is made formal by a technique from [HRS22] that shows that any sequence of $T$ instances generated by adaptive smoothed adversaries can be seen as a subset of $T/\sigma$ uniformly random instances from $\mathcal{X}$ with high probability. We implement this algorithmically by self-generating random instances and labels as a stand-in for the future. While the self-generated samples may not include adversary's next choice with some probability, these frameworks can be used to account for the uncertainty in each step. Furthermore, we show that the inclusion of additional self-generated samples has a small impact on the achievable regret by proving that the *regularized Rademacher complexity* (which acts as an admissible relaxation) is monotone in the set of generated samples (Lemma 4.1). This monotonicity property leads to regret bounds that gracefully degrade as a small function of $1/\sigma$.

**Stability, Poissonization and Generalization.** To obtain even stronger regret bounds for the binary setting, our analysis builds on the notion of *stability*, i.e., how little the distribution of learner's actions changes across time steps. A crucial ingredient in controlling stability is our novel Poissonization technique that randomly sets the *number of* samples, which are to be self-generated, from an appropriately chosen Poisson distribution. This allows us an additional degree of independence that is essential for controlling the loss from one step to the next using information theoretic techniques.

The stability analysis of the algorithm also depends crucially on a *modified generalization error* of the ERM, when it is trained on uniformly generated training samples and tested on smoothly distributed fresh instances. To bound this, we show a strong conditional independence property satisfied by the coupling from [HRS22]. This is instrumental for bounding the generalization error by allowing us to extract smooth variables from a set of uniform variables, which can then be used for symmetrization. We expect that this approach will be of independent interest for future work.

## 1.3 Related works

Our work relates to several paradigms and approaches to online learnability.

**Oracle-Efficient Online Learning.** Since the seminal work of [KV05] there has been a long line of work elucidating the computational aspects of online learning. [KV05] proposed the influential

follow-the-perturbed-leader algorithm. [KKL07] consider linear functions and study notions of regret when the learner is given access to an approximate optimization oracle. [KK05] study the transductive learning setting where instances (but not labels) are known in advance and give an oracle-efficient online algorithm. [RSS12] propose a general *relaxation* framework to develop efficient algorithms based on the upper bound of the value of the game. [DHL$^+$20] present the oracle-efficient Generalized-FTPL algorithm that is no-regret for a wide range of combinatorial and economics settings. On the flip size, [HK16] show that an $\Omega(\sqrt{N})$ lower bound is unavoidable in general in order to obtain nontrivial regret where the $N$ is the number of actions of the learner. This suggests that one needs to look beyond the worst-case in order to get truly efficient online algorithms.

**Beyond Worst-case Approaches to Online Learning.** Various notion of beyond worst-case behavior of online learning has been studied in the literature [RST11, HRS20, RS13b, DHJ$^+$17, BCKP20]. Most closely related are [RST11, HRS20, HRS22]. [RST11] studied smoothed analysis of online learning but only gave explicit regret bounds for simple classes such as thresholds. [HRS20, HRS22] both study the notion of smoothed analysis with adaptive adversary and show that statistically the regret is bounded by $O(\sqrt{Td\log(1/\sigma)})$ but do not provide efficient algorithms.

**Concurrent Work.** In a concurrent and independent work, [BDGR22] also gives oracle-efficient algorithms for smoothed online learning. In the binary classification setting, [BDGR22] obtains a regret bound of $\widetilde{O}(\sqrt{Td\sigma^{-1}})$ using an FTPL-based algorithm. In comparison, our result (Theorem 3.2) demonstrates a regret bound of $\widetilde{O}(\sqrt{Td\sigma^{-\frac{1}{2}}})$ with strictly better dependence on $\sigma$. Our regret bound's improved dependence on parameter $\sigma$ can be attributed to our novel technical innovations, including the introduction and careful analysis of *modified generalization error* and *stability* via a new coupling-based argument, and a Poissonization approach for *self-generating* samples that can leverage information theoretic arguments. For the case of real-valued functions with pseudo-dimension $d$, [BDGR22] achieve regret $\widetilde{O}(\sigma^{-1}\sqrt{Td})$ with $\widetilde{O}(\sqrt{T})$ calls to the oracle per round. In our paper, we obtain better regret of order $\widetilde{O}(\sqrt{Td\sigma^{-1}})$ using only 2 oracle calls per round [2]. Our stronger regret bounds are due to the fact that their algorithm is constrained to self-generating $T$-long sequences as a stand-in for the future, while we are generating substantially longer sequences that allow us to leverage the monotonicity of Rademacher complexity. Importantly, these novel techniques for achieving improved dependence on $\sigma$ also enable us to improve on the small-domain result of [DS16] and the transductive learning result of [KK05].

## 2 Preliminaries

### 2.1 Smoothed Online Learning

Let $\mathcal{X}$ be the space of instances, $\mathcal{Y} = [-1,1]$ be the space of labels, and $\mathcal{H} : \mathcal{X} \to \mathcal{Y}$ be the hypothesis class with pseudo dimension $d$ (See definition B.1 or [AB99] for the definition of pseudo dimension). Let $l : \mathcal{Y} \times \mathcal{Y} \to [0,1]$ be a convex loss function with Lipschitz constant $G$ in its first component. We also consider the special case where $\mathcal{Y} = \{-1, +1\}$ is binary and the hypothesis class $\mathcal{H}$ has VC dimension $d$. In this case, we consider the classification loss $l(\widehat{y}, y) = \mathbf{1}\{\widehat{y} \neq y\}$.

We work with the smoothed adaptive online adversarial setting from [HRS22]. We will consider $\sigma$-smooth adversaries, where a distribution is $\sigma$-smooth if its density is upper bounded by $1/\sigma$ times the density of the uniform distribution over the same domain. We remark that all of our results generalize to arbitrary known base distributions as well.

**Definition 2.1** ($\sigma$-smoothness). *Let $\mathcal{X}$ be a domain that supports a uniform distribution $\mathcal{U}$. A measure $\mu$ on $\mathcal{X}$ is $\sigma$-smooth if for all measurable subsets $A \subseteq \mathcal{X}$, $\mu(A) \leq \frac{\mathcal{U}(A)}{\sigma}$. The set of all $\sigma$-smooth distributions on domain $\mathcal{X}$ is denoted by $\Delta_\sigma(\mathcal{X})$.*

In online learning with adaptive smoothed adversaries, the learner and the adversary plays a repeated game for $T$ time steps. At each time step $t \in [T]$, the adversary chooses a $\sigma$-smooth distribution $\mathcal{D}_t^{\mathcal{X}} \in \Delta(\mathcal{X})$. A random instance $x_t \sim \mathcal{D}_t^{\mathcal{X}}$ is then drawn and presented to the learner. After receiving $x_t$, the learner predicts its label to be $\widehat{y}_t \in \mathcal{Y}$, while the adversary simultaneously chooses $y_t \in \mathcal{Y}$

---

[2]Subsequent to when the first versions of both papers were made available online, [BDGR22] improved their relaxation technique to achieve the same regret bound as ours, while still using more oracle calls.

as its true label. The learner then suffers loss $l(\widehat{y}_t, y_t)$. The above protocol is equivalent to a setting where the adversary chooses a distribution $\mathcal{D}_t \in \Delta(\mathcal{X} \times \mathcal{Y})$ over labeled instances $s_t = (x_t, y_t)$ whose marginal on $\mathcal{X}$ is $\sigma$-smooth, and the learner simultaneously chooses a classifier $h_t \in \mathcal{Y}^{\mathcal{X}}$. It is not hard to see that this equivalence holds for both randomized and deterministic interactions, see [RS14, Section 7.6] for a more detailed discussion. We will abbreviate $\mathcal{D}_t^{\mathcal{X}}$ to $\mathcal{D}_t$ when it is clear from the context.

We allow the adversary to be adaptive, i.e., the choice of $\mathcal{D}_t$ can depend on the realization of previous instances $\{(x_i, y_i)\}_{i=1}^{t-1}$ as well as the learner's previous predictions. We denote with $\mathscr{D}_\sigma$ the adaptive sequence of $\sigma$-smooth distributions $\mathcal{D}_1, \cdots, \mathcal{D}_T$ on the instances. Accordingly, let $\mathcal{Q}_t \in \Delta(\mathcal{Y})$ denote the learner's prediction rule on instance $x_t$, and let $\mathscr{Q}$ denote the adaptive sequence of distributions $\mathcal{Q}_1, \cdots, \mathcal{Q}_T$. We denote the expected regret of a learner with prediction rules $\mathscr{Q}$ on the adaptive sequence $\mathscr{D}_\sigma$ by

$$\mathbb{E}[\text{REGRET}(T, \mathscr{D}_\sigma, \mathscr{Q})] = \mathop{\mathbb{E}}_{\mathscr{D}_\sigma, \mathscr{Q}} \left[ \sum_{t=1}^{T} l(\widehat{y}_t, y_t) - \inf_{h \in \mathcal{H}} \sum_{t=1}^{T} l(h(x_t), y_t) \right].$$

We remove $\mathscr{D}_\sigma$ and $\mathscr{Q}$ from this notation when they are clear from the context.

An important property of smoothness is that it implies a probability coupling between uniform and adaptive smooth processes as first observed by [HRS22]. We will describe these in more details in Section 3 and 4.3. We use both the original result of [HRS22] (stated in Lemma B.1) and introduce a slightly strengthened version (in Lemma C.6).

## 2.2 Offline Optimization Oracle

We consider computationally efficient algorithms given access to an offline optimization oracle. For the case of binary classification, the oracle outputs the solution of empirical risk minimization on the input data.

**Definition 2.2** (ERM Oracle). *For a hypothesis class $\mathcal{H}$ and a loss function $l$, the oracle* $\mathsf{OPT}$ *(opt) takes a set [3] of inputs $S = \{(x_i, y_i)\}_{i \in [I]}$ where $(x_i, y_i) \in \mathcal{X} \times \mathcal{Y}$ for all $i \in [I]$ and returns*

$$\mathsf{OPT}_{\mathcal{H},l}(S) = \min_{h \in \mathcal{H}} \sum_{i=1}^{I} l(h(x_i), y_i) \ \text{ and } \ \mathsf{opt}_{\mathcal{H},l}(S) \in \arg\min_{h \in \mathcal{H}} \sum_{i=1}^{I} l(h(x_i), y_i).$$

For the case of real-valued functions, we consider an oracle that can minimize a mixture of binary and real-valued loss values defined below.

**Definition 2.3** (Real-valued optimization oracle). *For a hypothesis class $\mathcal{H}$ and two loss functions $l^r$ and $l^b$, the oracle* $\mathsf{OPT}$ *takes two sets of inputs $S$ and $S'$ over $\mathcal{X} \times \mathcal{Y}$ and returns*

$$\mathsf{OPT}_{\mathcal{H},l^r,l^b}(S; S') = \min_{h \in \mathcal{H}} \Big( \sum_{(x,y) \in S} l^{\mathrm{r}}(h(x), y) + \sum_{(x',y') \in S'} l^{\mathrm{b}}(h(x'), y') \Big).$$

We remark that these oracles are used in most previous works, including [RSS12]. They constitute a special form of regularized loss minimization oracles, where the regularization is given directly by a random process. For the binary setting where $\mathcal{Y} = \{\pm 1\}$ and $l^{\mathrm{r}} = l^{\mathrm{b}} = \mathbf{1}\{\widehat{y} \neq y\}$, the above optimization oracle is equivalent to ERM oracles.

We consider each call to the offline optimization oracle as having unit cost plus the additional runtime needed for creating and inputting the set of inputs that is linear in the length of the said histories. We note that our approach and results directly extend to using ERM oracles with (arbitrarily small) additive approximation error, such as those guaranteed by FPTAS optimization algorithms, using standard techniques presented by [DHL+20, Section 6].

**Remark 1.** *Though the oracles as defined above are required to work on arbitrary inputs, both Algorithms 1 and 2 from our work only call the optimization oracle on instances from the smoothed distributions or the uniform distribution. Thus, it suffices to have oracles that work for average-case instances. This makes the design of such oracles easier both from theoretical and practical points of view.*

---

[3] The inputs to the oracle are multisets. Unless specified otherwise, all the sets in this paper refer to multisets.

# 3 Oracle-Efficient Online Learning

## 3.1 Learning with Real-Valued Functions

In this section, we propose an oracle-efficient algorithm for real-valued functions with regret $\widetilde{O}(\sqrt{dT/\sigma})$. We consider the optimization oracle defined in Definition 2.3 with the loss functions specified by $l^{\mathrm{r}}(\hat{y}, y) = \frac{1}{2G} l(\hat{y}, y)$ and $l^{\mathrm{b}}(\hat{y}, y) = \mathbf{1}\{\hat{y} \neq y\} - \frac{1}{2}$.

We begin by describing our algorithm. At each time step $t \in [T]$, the algorithm draws $\widetilde{O}\left(\frac{T-t}{\sigma}\right)$ fresh new instances from the uniform distribution, denoted with $V^{(t)}$, together with their random labels $\mathcal{E}^{(t)}$ that are drawn i.i.d. from the Rademacher distribution, and treat them as hints for the future. Let $S^{(t)}$ denote the set of labeled instances $(V^{(t)}, \mathcal{E}^{(t)})$. Our algorithm then applies the offline optimization oracle to two input sequences: one where the real history $s_{1:t-1}$ is mixed with two copies[4] of $S^{(t)}$ and the current instance is labeled $+1$, and another, where the current label is labeled $-1$. Formally, we consider

$$\widehat{y}_t = \mathsf{OPT}\left(s_{1:t-1}; S^{(t)} \cup S^{(t)} \cup \{(x_t, -1)\}\right) - \mathsf{OPT}\left(s_{1:t-1}; S^{(t)} \cup S^{(t)} \cup \{(x_t, +1)\}\right). \quad (1)$$

Since the two input sequences to the optimization oracle only disagree on one label, the difference in the optimal errors is always within $[-1, +1]$, thus guarantees $\widehat{y}_t \in \mathcal{Y}$. Intuitively, $\hat{y}_t$ accounts for the gap between the errors of these two optimal classifiers so as to hedge the algorithm's bets against which instances will be played next. A formal description of the algorithm is given in Algorithm 1.

The main motivation for the algorithm is the coupling lemma of [HRS22]. It states that a sample from any $\sigma$-smooth distribution can be thought of as generated by first sampling $O\left(\sigma^{-1}\right)$ samples from the uniform distribution and then selecting one of them. The algorithm thus can be thought of as generating samples from the uniform distribution to account for the uncertainty in the adversary's choice. We will discuss this intuition and the proof of the following theorem in Section 4.3.

**Theorem 3.1** (Regret Upper Bound). *For any $\sigma$-smooth adversary $\mathscr{D}_\sigma$, Algorithm 1 has expected regret upper bounded by $\widetilde{O}(G\sqrt{Td/\sigma})$, where $\widetilde{O}$ hides factors that are polynomial in $\log(T)$ and $\log(1/\sigma)$. Here $G$ is the Lipschitz constant of the loss and $d$ is the pseudo-dimension of the class. Furthermore, the algorithm is oracle-efficient: at every round $t$, this algorithm uses two oracle calls with histories of length $\widetilde{O}(T/\sigma)$.*

---

**Algorithm 1:** Oracle-Efficient Smoothed Online Learning for Real-valued Functions

---
**Input:** $T, \sigma$
1  $K \leftarrow 100 \log T/\sigma$.
2  **for** $t \leftarrow 1$ **to** $T$ **do**
3  $\quad$ Receive $x_t$.
4  $\quad$ **for** $i = t+1, \cdots, T; \ k = 1, \cdots, K$ **do**
5  $\quad\quad$ Draw new $v_{i,k}^{(t)} \sim \mathcal{U}(\mathcal{X})$.
6  $\quad\quad$ Draw new $\epsilon_{i,k}^{(t)} \sim \mathcal{U}(\{-1, +1\})$.
7  $\quad$ **end**
8  $\quad S^{(t)} \leftarrow \left\{(v_{i,k}^{(t)}, \epsilon_{i,k}^{(t)})\right\}_{\substack{i=t+1:T \\ k=1:K}}$.
9  $\quad \widehat{y}_t \leftarrow \mathsf{OPT}\left(s_{1:t-1}; S^{(t)} \cup S^{(t)} \cup \{(x_t, -1)\}\right) - \mathsf{OPT}\left(s_{1:t-1}; S^{(t)} \cup S^{(t)} \cup \{(x_t, +1)\}\right)$.
10  $\quad$ Receive $y_t$, suffer loss $l(\widehat{y}_t, y_t)$.
11  **end**

---

## 3.2 Improved Bounds for Binary Classification

In this section, we focus on the important special case where the labels are binary and the loss function is the classification loss $\mathbf{1}\{\hat{y} \neq y\}$. We present Algorithm 2 that achieves regret $\widetilde{O}(\sqrt{Td\sigma^{-1/2}})$ with better dependence on the smoothness parameter $\sigma$ compared to Algorithm 1.

---

[4]We use two copies to scale the loss appropriately.

**Theorem 3.2** (Regret Bound for Efficient Smoothed Online Learning). *In the setting of binary classification with $\sigma$-smoothed adversaries, Algorithm 2 has expected regret that is at most*

$$\widetilde{O}\left(\min\left\{\sqrt{Td\sigma^{-1/2}}, \sqrt{T(d|\mathcal{X}|)^{1/2}}\right\}\right).$$

*Furthermore, Algorithm 2 is a* proper *learning oracle-efficient algorithm: at every round $t$, this algorithm uses a single ERM oracle call on a history that is of length $t + O(T/\sqrt{\sigma})$ with high probability.*

Unlike the algorithm for the real-valued case, the improved algorithm uses the FTPL framework. The algorithm itself is easy to describe: the algorithm generates a random number $N$ from the Poisson distribution with an appropriately chosen parameter, generates $N$ uniformly random points along with random labels and then predicts using the hypothesis that has the lowest error on the past data appended with the newly sampled data. On the surface, this algorithm seems similar to Algorithm 1. But there are two key differences: unlike Algorithm 1 which uses the difference in value of two optimizations, Algorithm 2 follows the prediction of the hypothesis with the lowest error. This makes Algorithm 2 a proper online learning algorithm. Secondly, unlike Algorithm 1 which uses a decreasing number of random examples over time, Algorithm 2 has the number of samples distributed according to a Poisson (with the same parameter in each step). The fact that the number of hints is drawn from the Poisson distribution is crucial for our analysis of the stability of the algorithm. We sketch the proof of the regret bounds in Section 4.4.

---

**Algorithm 2:** Smoothed Online Learning based on Poisson Number of Hints

**Input:** time horizon $T$, smoothness parameter $\sigma$, VC dimension $d$

1   $n \leftarrow \min\{T/\sqrt{\sigma}, T\sqrt{|\mathcal{X}|/d}\}$;

2   **for** $t \leftarrow 1$ **to** $T$ **do**

3      generate $N^{(t)} \sim \mathrm{Poi}(n)$ fresh hallucinated samples $(\widetilde{x}_1^{(t)}, \widetilde{y}_1^{(t)}), \cdots, (\widetilde{x}_N^{(t)}, \widetilde{y}_N^{(t)})$, which are
      i.i.d. conditioned on $N$ with $\widetilde{x}_i^{(t)} \sim \mathcal{U}(\mathcal{X})$ and $\widetilde{y}_i^{(t)} \sim \mathcal{U}(\{\pm 1\})$;

4      call the ERM oracle to compute $h_t \leftarrow \mathrm{opt}_{\mathcal{H},l}\left(\{(\widetilde{x}_i^{(t)}, \widetilde{y}_i^{(t)})\}_{i \in [N^{(t)}]} \cup \{x_\tau, y_\tau\}_{\tau \in [t-1]}\right)$;

5      observe $x_t$, predict $\widehat{y}_t = h_t(x_t)$, and receive $y_t$.

6 **end**

---

While our main interest is on beyond the worst-case adversaries, our results improve upon existing results for worst-case analysis of online learning as well. For finite domain and binary-valued loss settings where worst-case adversaries are vacuously $\sigma$-smooth for $1/\sigma = |\mathcal{X}|$, Theorem 3.2 also achieves an oracle-efficient regret bound of $O(\sqrt{T(d|\mathcal{X}|)^{1/2}})$, which is a refinement of $O(\sqrt{T|\mathcal{X}|})$ bound of [DS16], because VC dimension $d$ is at most $\mathcal{X}$, and is usually much smaller. Similarly, our bound can be instantiated in the setting of transductive learning with $|\mathcal{X}| = T$, which improves $O(T^{3/4}\sqrt{d})$ bound of [KK05] to $O(T^{3/4}d^{1/4})$.

**Corollary 3.3** (Regret for Small Domain and Transductive Learning). *There is an oracle-efficient algorithm for online learning with binary labels (in the worst-case) that achieves an expected regret of $O(\sqrt{T(d|\mathcal{X}|)^{1/2}})$ for any hypothesis class with VC dimension $d$ on domain $\mathcal{X}$. For transductive learning with binary labels, there is an oracle-efficient algorithm, with expected regret $O\left(T^{3/4}d^{1/4}\right)$.*

**Remark 2.** *Both Algorithms 1 and 2 can be adapted to deal with unknown $\sigma$. In particular, the regret bounds hold for any approximation $\widetilde{\sigma}$ that is a lower bound of the real $\sigma$ up to constant multiplicative factors. This corresponds to settings where the world is more smooth than we give it credit. Even when we have extremely poor upper and lower bounds, we can use hedging to still get non-trivial regret with only a minor blow up in computation. We will provide more details in Appendix D about working with knowledge of approximate $\widetilde{\sigma}$.*

## 4   Proof Sketches for Main Regret Bounds

Before discussing the proof sketches for the main regret bounds, we will first introduce two frameworks for designing efficient algorithms for online learning.

## 4.1 Relaxations and Admissibility

The proof of Theorem 3.1 relies on the *admissible relaxation* framework proposed in [RSS12]. A relaxation $\mathbf{Rel}_T$ is a sequence of functions $\mathbf{Rel}_T(\mathcal{H}|s_{1:t})$ for each $t \in [T]$, which map the history of the play to real values that upper bounds the conditional value of the game. We will make use of an important algorithmic aspect of the relaxation framework, which states that whenever an algorithm is *admissible* with respect to some relaxation, its expected regret can be upper bounded in terms of the value of the relaxation at the beginning of the game.

**Definition 4.1** (Admissibility). *In the smoothed online learning setting, let $\mathscr{Q}$ be an algorithm that gives rise to a sequence of distributions $\mathcal{Q}_1, \cdots, \mathcal{Q}_T$ on the predicted labels. We say $\mathscr{Q}$ is admissible with respect to a relaxation $\{\mathbf{Rel}_T(\mathcal{H} \mid s_{1:t})\}_{t=0}^T$, if for any sequence of instances $s_{1:T}$,*

*1. For all $t \in [T]$,*

$$\sup_{\mathcal{D}_t \in \Delta_\sigma(\mathcal{X})} \mathbb{E}_{x_t \sim \mathcal{D}_t} \sup_{y_t \in \mathcal{Y}} \left\{ \mathbb{E}_{\widehat{y}_t \sim \mathcal{Q}_t}[l(\widehat{y}_t, y_t)] + \mathbf{Rel}_T(\mathcal{H} \mid s_{1:t-1} \cup (x_t, y_t)) \right\} \leq \mathbf{Rel}_T(\mathcal{H} \mid s_{1:t-1}),$$

*where $\Delta_\sigma(\mathcal{X})$ is the set of $\sigma$-smooth distributions on $\mathcal{X}$;*
*2. The final value satisfies $\mathbf{Rel}_T(\mathcal{H} \mid s_{1:T}) \geq -\inf_{h \in \mathcal{H}} L(h, s_{1:T})$.*

The following proposition is the analog of the results of [RSS12] when the adversary is smooth. The full proof is presented in Appendix G.

**Proposition 4.1** (Regret Bound via Admissibility). *In the smoothed online learning setting, let $\mathscr{Q} = (\mathcal{Q}_1, \cdots, \mathcal{Q}_T)$ be an algorithm that is admissible with respect to relaxations $\mathbf{Rel}_T(\mathcal{H})$, then the following bound on the expected regret holds regardless of the strategies $\mathscr{D}_\sigma$ of the adversary:*

$$\mathbb{E}[\textsc{Regret}(T, \mathscr{Q}, \mathscr{D}_\sigma)] \leq \mathbf{Rel}_T(\mathcal{H} \mid \emptyset) + O(\sqrt{T}).$$

## 4.2 Follow the Perturbed Leader

When the labels are binary, Algorithm 2 achieves an improved regret using the Follow the Perturbed Leader (FTPL) principle [KV05]. An FTPL algorithm makes predictions by applying ERM oracle to the perturbed histories of the play. At every time step $t \in [T]$, the algorithm chooses a distribution over labeled instances, from which it draws $N$ random instances $(\widetilde{x}_1^{(t)}, \widetilde{y}_1^{(t)}), \cdots, (\widetilde{x}_N^{(t)}, \widetilde{y}_N^{(t)})$. The predicted label is then given by $\widehat{y}_t = h_t(x_t)$, where

$$h_t \leftarrow \mathsf{opt}_{\mathcal{H}, l}\left( s_{1:t-1} \cup \{(\widetilde{x}_i^{(t)}, \widetilde{y}_i^{(t)})\}_{i \in [N]} \right).$$

The standard analysis of FTPL bounds the expected regret as follows:

$$\mathbb{E}[\textsc{Regret}] \leq \underbrace{\mathbb{E}\left[ \sum_{t=1}^T l(h_t(x_t), y_t) - l(h_{t+1}(x_t), y_t) \right]}_{\text{Stability}} + \underbrace{\mathbb{E}\left[ \sup_{h \in \mathcal{H}} \sum_{i=1}^N l(h(\widetilde{x}_i), \widetilde{y}_i) - \sum_{i=1}^N l(h^*(\widetilde{x}_i), \widetilde{y}_i) \right]}_{\text{Perturbation}},$$

where $h^* = \arg\inf_{h \in \mathcal{H}} \sum_{t=1}^T l(h(x_t), y_t)$.

Note that the perturbation term is already well-understood from statistical learning theory since it is essentially the Rademacher complexity of $\mathcal{H}$ for sample size $N$. Therefore, we will focus on bounding the stability term by designing perturbations that can leverage the anti-concentration property of smoothed adversaries.

## 4.3 Proof Sketch of Theorem 3.1

Elaborating on the intuition laid out in Section 3, we will use the coupling technique introduced by [HRS22] (see Lemma B.1 for a complete description) to replace the sequence of $T$ random inputs $\{x_1, \cdots, x_T\}$ generated by the adaptive adversary with $TK$ inputs $\{z_{t,k}\}_{t \in [T], k \in [K]}$ that are generated i.i.d. from the uniform distribution over $\mathcal{X}$, such that with high probability $\{x_1, \cdots, x_T\} \subseteq \{z_{t,k}\}_{t \in [T], k \in [K]}$. This implies that, up to a small probability of failure, it is sufficient to consider a simpler setting where the adversary is promised to pick future instances from a larger set of uniformly distributed samples (which we call *the set of hints*). This setting differs from the standard transductive

learning setting [KK05, CbS11] in two significant ways: 1) The set of hints is not revealed to the learner beforehand; 2) the hint set is larger, by a multiplicative factor of $K \approx 1/\sigma$, than the set of realized instances.

It turns out that both issues can be handled elegantly in the admissible relaxations framework of [RSS12]. For the first issue, note that the Algorithm 1 can be seen as self-generating hints and although they do not necessarily correspond to the adversary's sample, the relaxation-based argument guarantees that matching the randomness of hints at a distribution level suffices to bound the regret of the algorithm (see Appendix B.7 for more details). For the second issue, our relaxation will be based on a characterization of the uncertainty in the future that is *monotone* in the set of hints, which we call *regularized Rademacher complexity*. Formally, for a set of unlabeled instances $Z = \{z_i\}_{i=1}^{I}$ and a function $\Phi : \mathcal{H} \to \mathbb{R}$, the Rademacher complexity for set $Z$ regularized by $\Phi$ is defined as

$$\mathfrak{R}(\Phi, Z) = \mathop{\mathbb{E}}_{\epsilon_{1:I} \overset{\text{iid}}{\sim} \mathcal{U}(\pm 1)} \left[ \sup_{h \in \mathcal{H}} \left\{ \sum_{i \leq I} \epsilon_i h(z_i) + \Phi(h) \right\} \right].$$

We show that regularized Rademacher complexity is monotone as a function of the dataset. See Appendix B.2 for a proof of Lemma 4.1.

**Lemma 4.1** (Monotonicity of Regularized Rademacher Complexity). *For any dataset $z_{1:m} \in \mathcal{X}^m$ and any additional data point $x \in \mathcal{X}$, we have $\mathfrak{R}(\Phi, z_{1:m}) \leq \mathfrak{R}(\Phi, z_{1:m} \cup \{x\})$.*

This monotonicity ensures that using the hint set, which is a superset of possible instances, will still lead to a no-regret algorithm.

Finally, the relaxation we use to analyze Algorithm 1 is the expected Rademacher complexity of the union of future hints, regularized by the past total loss, i.e.,

$$\mathbf{Rel}_T(\mathcal{H} \mid s_{1:t}) = 2G \mathop{\mathbb{E}}_{V^{(t)} \overset{\text{iid}}{\sim} \mathcal{U}(\mathcal{X})} \left[ \mathfrak{R}(-L^{\text{r}}(\cdot, s_{1:t}), V^{(t)}) \right] + 2G\beta(T - t),$$

where $L^{\text{r}}(h, s_{1:t}) = \sum_{i=1}^{t} l^{\text{r}}(h(x_i), y_i)$ for $h \in \mathcal{H}$. Here $\beta = 10TK(1 - \sigma)^K \in o(1/T)$ represents the penalty caused by the failure of coupling. Once admissibility is established, we will obtain a regret bound using Proposition 4.1. See Appendix B for a complete proof of the theorem. We remark that the final regret bound can also be stated in terms of the Rademacher complexity which is data dependent and adapts readily to benign structures of the instance domain.

## 4.4 Proof Sketch of Theorem 3.2

Let $\mathcal{Q}_t$ be the distribution of the learner's action $h_t \in \mathcal{H}$ in Algorithm 2 and $\mathcal{D}_t$ denote the distribution of the adversary at time $t$. The main quantity we will use to analyze the algorithm is the stability

$$\text{Stability} = \mathop{\mathbb{E}}_{s_t \sim \mathcal{D}_t} \left( \mathop{\mathbb{E}}_{h_t \sim \mathcal{Q}_t} [L(h_t, s_t)] - \mathop{\mathbb{E}}_{h_{t+1} \sim \mathcal{Q}_{t+1}} [L(h_{t+1}, s_t)] \right).$$

We analyze this expression by breaking it down into a sum of two quantities:

$$\text{Stability} \leq \text{TV}(\mathcal{Q}_t, \mathop{\mathbb{E}}_{s_t \sim \mathcal{D}_t} [\mathcal{Q}_{t+1}]) + \underbrace{\mathop{\mathbb{E}}_{s_t, s'_t \sim \mathcal{D}_t; R^{(t+1)}} [L(h_{t+1}, s'_t) - L(h_{t+1}, s_t)]}_{\text{Modified generalization error}}.$$

Here, $R^{(t)}$ is the fresh randomness generated by the algorithm at the beginning of time $t$.

In order to see where this expression comes from, note that the first term itself would be an upper bound on the stability, if neither of $\mathcal{Q}_t$ and $\mathcal{Q}_{t+1}$ depend on the new observation $s_t = (x_t, y_t)$ at time $t$. However, while $\mathcal{Q}_t$ is independent of $s_t$, $\mathcal{Q}_{t+1}$ does depend on $s_t$ because $h_{t+1}$ is trained on $s_t$. To overcome this dependence, we introduce a ghost sample $s'_t$ that allows us to decouple $h_{t+1} \sim \mathcal{Q}_{t+1}$ and the new observation. This gives rise to the second term which we call *modified generalization error*. We formally discuss this decomposition in Appendix C.2.

The first term is the total variation (TV) distance between $\mathcal{Q}_t$ and the mixture distribution $\mathbb{E}_{s_t \sim \mathcal{D}_t}[\mathcal{Q}_{t+1}]$. In order to bound this term, we closely use the independence properties of the Poisson distribution which allows us to write an explicit expression for the total variation distance which we can then bound using the Ingster–Suslina method. We formally prove this in Lemma C.4.

The intuitive idea behind bounding the second term is as follows. Consider the simpler setting of $t = 1$ (i.e. no history) and $\mathcal{D}_t = \mathcal{U}(\mathcal{X} \times \{\pm 1\})$ (i.e. the new observation $s_t$ follows the same distribution as the self-generated samples). In this case, the generalization error is precisely the difference between the test error and the training error with $N + 1$ iid training data, and classical Rademacher complexity gives an upper bound $O(\sqrt{d/N})$. For general $\sigma$-smooth $\mathcal{D}_t$, we establish a strong conditional independence property of the coupling argument from [HRS22]. This states that there exists a coupling between uniform and adaptive smooth processes, such that when the inclusion property is satisfied, the distribution of the realized uniform variables *conditional on the unrealized uniform variables* is also identical to the smooth distributions given by the adversary. This will be instrumental for bounding the generalization error by allowing us to extract smooth variables from a set of uniform variables, which can then be used to for the purpose of symmetrization. We formally prove the upper bound on the modified generalization in Lemma C.5.

Using this bound on stability, we can get a bound on the regret of the algorithm using analysis techniques for FTPL. For a full proof, see Appendix C.

**Remark 3.** *The proof for the modified generalization needs the smoothness of both the covariates $x$ and labels $y$. This can be ensured with a loss of a constant in the binary (and generally the finite label space) setting. This is the main reason that this proof does not generalize to the real valued setting.*

## 5    Discussion, Additional Results, and Open Problems

**Computational Lower Bounds.**    Our main contribution is oracle-efficient online learning algorithms in the smoothed setting that achieve an $O(\sqrt{dT\sigma^{-1}})$ regret upper bound in the real-valued case, and an $O(\sqrt{dT\sigma^{-1/2}})$ upper bound for binary classification. However, neither of these upper bounds is statistically optimal; the statistically optimal regret here is $\widetilde{\Theta}(\sqrt{dT\log(1/\sigma)})$ [HRS22]. We ask the following question: is the above discrepancy an artifact of our regret analysis, or an intrinsic limitation of our or all oracle-efficient algorithms.

We show in Theorem E.1 that the regret of our current family of algorithms cannot be improved by tuning parameters. In addition, we also show the following general lower bound for computationally efficient algorithms. See Appendix E for the proofs and more details.

**Theorem 5.1** (Computational Lower Bound for Smoothed Online Learning)**.** *For $1/\sigma \geq d$, any proper algorithm which only has access to the ERM oracle and achieves a regret $o(\min\{T, \sqrt{T(d/\sigma)^{1/2}}\})$ for any $\sigma$-smoothed online learning problem must have an $\omega(\sqrt{d/\sigma})$ total running time. Here, the total running time refers to the total number of operations performed by our algorithm, in which each oracle call takes unit time, and maintaining each element in the input to the oracle also takes unit time.*

Theorem 5.1 implies an exponential statistical-computational gap in smoothed online learning: for exponentially small $\sigma$, achieving the statistical regret $\widetilde{O}(\sqrt{Td\log(1/\sigma)})$ requires an exponential running time. However, Theorem 5.1 still exhibits gaps to our computational upper bounds. We discuss this further in the appendix and present the following open problem.

**Open Question.** *For $d/\sigma \gg T^2$ in the smoothed setting, does any algorithm achieving $o(T)$ regret require $\Omega(\mathrm{poly}(T, 2^d, 1/\sigma))$ computational time given access to the ERM oracle?*

**Other Constrained Adversary Models**    Online learning in presence of constrained adversaries is an active research area with several parallel proposed models. While the main focus of our work is on the smoothed analysis framework, our techniques extend to several other frameworks and establish a connection between existing lines of work. In particular, our application of the probability coupling technique to hallucinate a superset of future instances relates to an extension of the framework of transductive learning (and the literature on hints) to a setting where the learner has access to $K$ hints per time step in the future. We call this the $K$-*hints transductive* framework and formally address it in Appendix H. More broadly, smoothed analysis also related to a line of work on *predictable sequences* as it enforces a notion of similarity between future instances that holds at the stochastic level. We believe exploring the connections between these various models of constrained and predictable adversaries in online learning is an important research direction for the future.

## Acknowledgments and Disclosure of Funding

This work was supported in part by the National Science Foundation under grant CCF-2145898, a C3.AI Digital Transformation Institute grant, and Berkeley AI Research Commons grants. This work was partially done while authors were visitors at the Simons Institute for the Theory of Computing.

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

# 6 Check List

