# A  Additional Related Work

**Beyond Worst-Case Online Learning**   [RST11] consider online learning where the adversary is constrained and build a framework based on minimax analysis and constrained sequential Rademacher complexity to analyze regret in these scenarios. These techniques have been applied to other constrained settings [KAH$^+$19].

**Smoothed Online Learning**   [GR17] consider smoothed online learning when looking at problems in online algorithm design. They prove that while optimizing parameterized greedy heuristics for combinatorial problems in presence of smoothing this problem can be learned with non-trivial sublinear regret. [CAK17] consider the same problem with an emphasis on the per-step runtime being logarithmic in $T$. [HRS20, HRS22] both study the notion of smoothed analysis with adaptive adversary and show that statistically the regret is bounded by $O(\sqrt{Td\log(1/\sigma)})$.

**Online Learning with Predictable Sequences**   Another line of work has focused on the future sequences being predictable given the past instances. [RS13b] incorporate additional information available in terms of an estimator for future instances. They achieve regret bounds depending on the path length of these estimators and can beat the worst-case $\Omega(\sqrt{T})$ if the estimators are accurate. [HM07] models predictability as knowing the first coordinate of loss vectors, which is revealed to the learner before he chooses actions. Some other work model predictability through hints which are additive estimate of loss vectors [HK10, RS13a, SL14, MY16]. [DHJ$^+$17, BCKP20] considers settings where the learner has access to hints in form of vectors that are weakly correlated with the future instances and show exponential improvement in the regret in some cases. The literature on hints represents an active and growing subarea of online learning (see [BCKP20] and references within).

# B  Oracle-Efficient Learning with Real-valued Functions

## B.1  Coupling Lemma

**Lemma B.1** (Coupling, [HRS22]). *Let $\mathscr{D}_\sigma$ be an adaptive sequence of $t$ $\sigma$-smooth distributions on $\mathcal{X}$. Then, there is a coupling $\Pi$ such that $\left(x_1, z_{1,1}, \ldots, z_{1,K}, \ldots, x_t, z_{t,1}, \ldots, z_{t,K}\right) \sim \Pi$ satisfy*

   *a.  $x_1, \ldots, x_t$ is distributed according $\mathscr{D}_\sigma$.*

   *b.  For every $j \leq t$, $\{z_{i,k}\}_{i \geq j, k \in [K]}$ are uniformly and independently distributed on $\mathcal{X}$, conditioned on $x_1, \ldots, x_{j-1}$.*

   *c.  With probability at least $1 - t\left(1-\sigma\right)^K$, $\{x_1, \cdots, x_T\} \subseteq \{z_{t,k}\}_{t=1:T, k=1:K}$.*

## B.2  Monotonicity of the Regularized Rademacher Complexity

**Lemma 4.1** (Restated). *Let $Z = \{z_i\}_{i \in [m]} \in \mathcal{X}^m$ be a set of unlabeled instances and $\Phi : \mathcal{H} \to \mathbb{R}$ be a mapping from the set of hypothesis to real values. Recall that the Rademacher complexity for set $Z$ regularized by $\Phi$ is defined as*

$$\mathfrak{R}(\Phi, Z) = \mathop{\mathbb{E}}_{\epsilon_{1:m} \overset{iid}{\sim} \mathcal{U}(\pm 1)} \left[ \sup_{h \in \mathcal{H}} \left\{ \sum_{i=1}^m \epsilon_i h(z_i) + \Phi(h) \right\} \right].$$

*Then for any dataset $z_{1:m} \in \mathcal{X}^m$ and any additional data point $x \in \mathcal{X}$, we have*

$$\mathfrak{R}(\Phi, z_{1:m}) \leq \mathfrak{R}(\Phi, z_{1:m} \cup \{x\}).$$

*Proof of Lemma 4.1.* Using $\mathbb{E}[\sup_\lambda X_\lambda] \geq \sup_\lambda \mathbb{E}[X_\lambda]$, we have

$$\Re(\Phi, Z \cup \{x\}) = \mathop{\mathbb{E}}_{\epsilon_{1:m+1}} \left[ \sup_{h \in \mathcal{H}} \left\{ \sum_{i=1}^{m} \epsilon_i h(z_i) + \epsilon_{m+1} h(x) + \Phi(h) \right\} \right]$$

$$= \mathop{\mathbb{E}}_{\epsilon_{1:m}, \epsilon_{m+1}} \left[ \sup_{h \in \mathcal{H}} \left\{ \sum_{i=1}^{m} \epsilon_i h(z_i) + \epsilon_{m+1} h(x) + \Phi(h) \right\} \right]$$

$$\geq \mathop{\mathbb{E}}_{\epsilon_{1:m}} \left[ \sup_{h \in \mathcal{H}} \left\{ \sum_{i=1}^{m} \epsilon_i h(z_i) + \mathop{\mathbb{E}}_{\epsilon_{m+1}} [\epsilon_{m+1} h(x)] + \Phi(h) \right\} \right]$$

$$= \Re(\Phi, Z),$$

as desired. $\qquad\square$

## B.3  Notions for Real-Valued Functions

In this section we introduce the notions that will be useful in analyzing real-valued hypothesis classes, including pseudo dimension and covering numbers.

**Definition B.1** (Pseudo-dimension, [AB99])**.** *For every $h \in \mathcal{H}$, let $B_h(x, y) = \text{sgn}(h(x) - y)$ be the indicator of the region below or on the graph of $h$. The pseudo-dimension of hypothesis class $\mathcal{H}$ is defined as the VC dimension of the subgraph class $B_{\mathcal{H}} = \{B_h : h \in \mathcal{H}\}$.*

We will see in the two following lemmas that pseudo dimension can be used to characterize the magnitude of covering numbers and Rademacher complexity.

**Lemma B.2** ($d_{L_1(\mathcal{U})}$-Covering Number Bound, [AB99])**.** *The $\epsilon$-covering number of $\mathcal{H}$ with respect to metric $d_{L_1(\mathcal{U})}$, denoted by $\mathcal{N}(\epsilon, \mathcal{H}, L_1(\mathcal{U}(\mathcal{X})))$, is the cardinality of the smallest subset $\mathcal{H}'$ of $\mathcal{H}$, such that for every $h \in \mathcal{H}$, there exists $h' \in \mathcal{H}'$ such that $d_{L_1(\mathcal{U})}(h, h') \leq \epsilon$, where $d_{L_1(\mathcal{U})}(f, g) = \mathbb{E}_{\mathcal{U}}[|f - g|]$. If $d$ is the pseudo-dimension of $\mathcal{H}$, then for any $\epsilon > 0$,*

$$\log \mathcal{N}(\epsilon, \mathcal{H}, L_1(\mathcal{U}(\mathcal{X}))) \in \widetilde{O}\left( d \log\left(\frac{1}{\epsilon}\right) \right).$$

**Lemma B.3** (Rademacher Complexity Bound, [Bar06])**.** *The Rademacher complexity of class $\mathcal{H}$ for a set of $n$ elements is upper bounded by $O\left(\sqrt{dn \log n}\right)$, where $d$ is the pseudo dimension of $\mathcal{H}$.*

## B.4  Proof of Theorem 3.1

**Theorem 3.1** (Restated)**.** *For any $\sigma$-smooth adversary $\mathscr{D}_\sigma$, Algorithm 1 has expected regret upper bounded by $\widetilde{O}(G\sqrt{Td/\sigma})$, where $\widetilde{O}$ hide factors that are polynomial in $\log(T)$ and $\log(1/\sigma)$. Here $G$ is the Lipschitz constant of the loss and $d$ is the pseudodimension of class $\mathcal{H}$. Furthermore, the algorithm is oracle-efficient: at every round $t$, this algorithm uses two oracle calls with histories of length $\widetilde{O}(T/\sigma)$.*

*Proof.* To prove Theorem 3.1 we use the following relaxation:

$$\mathbf{Rel}_T(\mathcal{H}|s_{1:t}) = 2G \mathop{\mathbb{E}}_{V^{(t)} \overset{\text{iid}}{\sim} \mathcal{U}(\mathcal{X})} \left[ \Re(-L^{\text{r}}(\cdot, s_{1:t}), V^{(t)}) \right] + 2G\beta(T - t)$$

$$= 2G \mathop{\mathbb{E}}_{V^{(t)}, \mathcal{E}^{(t)}} \left[ \sup_{h \in \mathcal{H}} \left\{ \sum_{\substack{i=t+1:T \\ k=1:K}} \epsilon_{i,k}^{(t)} h(v_{i,k}^{(t)}) - L^{\text{r}}(h, s_{1:t}) \right\} \right] + 2G\beta(T - t), \quad (2)$$

where $K = 100 \log T / \sigma$ and $\beta = 10TK(1-\sigma)^K$. We will show in Lemma B.4 that the above relaxation is admissible. Therefore, Proposition 4.1 gives us the following upper bound on the expected regret:

$$\mathbb{E}[\text{REGRET}(T)] \leq \mathbf{Rel}_T(\mathcal{H}|\emptyset) + O(\sqrt{T}) = 2G \underbrace{\mathop{\mathbb{E}}_{V^{(0)}, \mathcal{E}^{(0)}}\left[\sup_{h \in \mathcal{H}}\left\{\sum_{\substack{i=1:T \\ k=1:K}} \epsilon_{i,k}^{(0)} h(v_{i,k}^{(0)})\right\}\right]}_{(a)} + 2G\beta T + O(\sqrt{T}).$$

The first term (a) is the Rademacher complexity of the hypothesis class $\mathcal{H}$ with respect to the uniform distribution for sample size $TK$. By Lemma B.3, $(a) \leq O\left(\sqrt{dTK \log(TK)}\right)$. For the second term, we have $\beta T \in o(1)$ because $\beta \lesssim TKe^{-\sigma K} \lesssim T^{-99} \log T / \sigma = o(1/T)$. Plugging in $K = O\left(\log(T)/\sigma\right)$, we have the following bound:

$$\mathbb{E}[\text{REGRET}(T)] \leq O\left(G\sqrt{\frac{dT}{\sigma} \log T \log\left(\frac{T}{\sigma}\right)}\right) \subseteq \widetilde{O}(G\sqrt{dT/\sigma}),$$

where $\widetilde{O}$ hide factors that are polynomial in $\log(T)$ and $\log(1/\sigma)$. $\qquad\square$

## B.5 Admissibility of the Relaxation

**Lemma B.4.** *The prediction rule $\mathcal{Q} = (\mathcal{Q}_1, \cdots, \mathcal{Q}_T)$ given by Algorithm 1 is admissible with respect to the relaxation defined in Equation* (2).

*Proof.* Using the language of regularized Rademacher complexity, the above relaxation can be written as

$$\mathbf{Rel}_T(\mathcal{H}|s_{1:t}) = 2G \mathop{\mathbb{E}}_{V^{(t)} \overset{\text{iid}}{\sim} \mathcal{U}(\mathcal{X})}\left[\mathfrak{R}(-L^{\text{r}}(\cdot, s_{1:t}), V^{(t)})\right] + 2G\beta(T-t),$$

where $L^{\text{r}}(\cdot, s_{1:t}) = \sum_{i=1}^{t-1} l^{\text{r}}(h(x_i), y_i)$. When $t = T$, the relaxation becomes

$$\mathbf{Rel}_T(\mathcal{H}|s_{1:T}) = -2GL^{\text{r}}(h, s_{1:T}) = -\inf_{h \in \mathcal{H}} \sum_{i=1}^{T} l(h(x_i), y_i),$$

thus it satisfies the second condition of Definition 4.1. For the first condition, we need to verify

$$\sup_{\mathcal{D}_t \in \mathfrak{D}_t} \mathop{\mathbb{E}}_{x_t \sim \mathcal{D}_t} \sup_{y_t \in \mathcal{Y}} \left\{\mathop{\mathbb{E}}_{\widehat{y}_t \sim \mathcal{Q}_t}[l(\widehat{y}_t, y_t)] + \mathbf{Rel}_T(\mathcal{H} \mid s_{1:t-1} \cup (x_t, y_t))\right\} \leq \mathbf{Rel}_T(\mathcal{H} \mid s_{1:t-1}). \quad (3)$$

We first upper bound the LHS of Equation (3) by matching the randomness in $V^{(t)}$ and applying Jensen's inequality to the supremum function. For every fixed input $x_t$ and hint set $V^{(t)}$, we denote our prediction rule in Equation (1) with $\mathcal{Q}_t(V^{(t)})$. This gives us

$$\sup_{\mathcal{D}_t \in \mathfrak{D}_t} \mathop{\mathbb{E}}_{x_t \sim \mathcal{D}_t} \sup_{y_t \in \mathcal{Y}} \left\{\mathop{\mathbb{E}}_{\widehat{y}_t \sim \mathcal{Q}_t}[l(\widehat{y}_t, y_t)] + \mathbf{Rel}_T(\mathcal{H} \mid s_{1:t-1} \cup (x_t, y_t))\right\}$$

$$= \sup_{\mathcal{D}_t \in \mathfrak{D}_t} \mathop{\mathbb{E}}_{x_t \sim \mathcal{D}_t} \sup_{y_t \in \mathcal{Y}} \mathop{\mathbb{E}}_{V^{(t)} \overset{\text{iid}}{\sim} \mathcal{U}(\mathcal{X})}\left[\mathop{\mathbb{E}}_{\widehat{y}_t \sim \mathcal{Q}_t(V^{(t)})}\left[l(\widehat{y}_t, y_t)\right] + 2G \cdot \mathfrak{R}(-L^{\text{r}}(\cdot, s_{1:t}), V^{(t)})\right] + 2G\beta(T-t) \quad (4)$$

$$\leq \sup_{\mathcal{D}_t \in \mathfrak{D}_t} \mathop{\mathbb{E}}_{x_t \sim \mathcal{D}_t} \mathop{\mathbb{E}}_{V^{(t)} \overset{\text{iid}}{\sim} \mathcal{U}(\mathcal{X})}\left[\sup_{y_t \in \mathcal{Y}}\left\{\mathop{\mathbb{E}}_{\widehat{y}_t \sim \mathcal{Q}_t(V^{(t)})}\left[l(\widehat{y}_t, y_t)\right] + 2G \cdot \mathfrak{R}(-L^{\text{r}}(\cdot, s_{1:t}), V^{(t)})\right\}\right] + 2G\beta(T-t) \quad (5)$$

In Equation (5), note that $\mathcal{Q}_t(V^{(t)})$ is the same as the transductive prediction rule in [RSS12, Equation (25)], with $V^{(t)}$ being the set of unlabeled future instances and $s_{1:t-1}$ being the historical data with labels.

According to [RSS12, Lemma 12], for all input $x_t$ and unlabeled sequence $\mathscr{X}$ (which plays the role of $x_{t+1:T}$), the decision rule $\mathcal{Q}_t(\mathscr{X})$ satisfies

$$\sup_{y_t \in \mathcal{Y}} \left\{ \mathop{\mathbb{E}}_{\widehat{y}_t \sim \mathcal{Q}_t(\mathscr{X})} [l(\widehat{y}_t, y_t)] + 2G \mathop{\mathbb{E}}_{\mathcal{E}} \left[ \sup_{h \in \mathcal{H}} \left\{ \sum_{x \in \mathscr{X}} \epsilon_x h(x) - L^{\mathrm{r}}(h, s_{1:t-1} \cup (x_t, y_t)) \right\} \right] \right\}$$

$$\leq 2G \mathop{\mathbb{E}}_{\mathcal{E}} \left[ \sup_{h \in \mathcal{H}} \left\{ \sum_{x \in \mathscr{X} \cup \{x_t\}} \epsilon_x h(x) - L^{\mathrm{r}}(h, s_{1:t-1}) \right\} \right].$$

Therefore, if we choose the sequence $\mathscr{X}$ to be $V^{(t)}$, we obtain the following inequality which is written in the language of regularized Rademacher complexity:

$$\sup_{y_t \in \mathcal{Y}} \left\{ \mathop{\mathbb{E}}_{\widehat{y}_t \sim \mathcal{Q}_t(V^{(t)})} \left[ l(\widehat{y}_t, y_t) \right] + 2G \cdot \mathfrak{R}(-L^{\mathrm{r}}(\cdot, s_{1:t}), V^{(t)}) \right\} \leq 2G \cdot \mathfrak{R}(-L^{\mathrm{r}}(\cdot, s_{1:t-1}), V^{(t)} \cup \{x_t\}).$$

By adding the expectations over $V^{(t)}$ and $x_t$ on both sides, we obtain the following upper bound:

$$(5) \leq \sup_{\mathcal{D}_t \in \mathfrak{D}_t} \mathop{\mathbb{E}}_{x_t \sim \mathcal{D}_t} \mathop{\mathbb{E}}_{V^{(t)} \overset{\mathrm{iid}}{\sim} \mathcal{U}(\mathcal{X})} \left[ 2G \cdot \mathfrak{R}(-L^{\mathrm{r}}(\cdot, s_{1:t-1}), V^{(t)} \cup \{x_t\}) \right] + 2G\beta(T-t)$$

$$\leq \mathop{\mathbb{E}}_{V^{(t)} \overset{\mathrm{iid}}{\sim} \mathcal{U}(\mathcal{X})} \left[ \sup_{\mathcal{D}_t \in \mathfrak{D}_t} \mathop{\mathbb{E}}_{x_t \sim \mathcal{D}_t} 2G \cdot \mathfrak{R}(-L^{\mathrm{r}}(\cdot, s_{1:t-1}), V^{(t)} \cup \{x_t\}) \right] + 2G\beta(T-t).$$

According to Lemma B.5, we can replace the $x_t$ sampled from the worst-case smooth distribution by $Z_t$ sampled independently from the uniform distribution, with the extra cost $\beta$. This gives

$$(5) \leq \mathop{\mathbb{E}}_{V^{(t)} \overset{\mathrm{iid}}{\sim} \mathcal{U}(\mathcal{X})} \mathop{\mathbb{E}}_{Z_t \overset{\mathrm{iid}}{\sim} \mathcal{U}(\mathcal{X})} \left[ 2G \cdot \left( \mathfrak{R}(-L^{\mathrm{r}}(\cdot, s_{1:t-1}), V^{(t)} \cup Z_t) + \beta \right) \right] + 2G\beta(T-t)$$

$$= 2G \mathop{\mathbb{E}}_{V^{(t)} \overset{\mathrm{iid}}{\sim} \mathcal{U}(\mathcal{X})} \left[ \mathfrak{R}(-L^{\mathrm{r}}(\cdot, s_{1:t-1}), V^{(t-1)}) \right] + 2G\beta(T-t+1) \tag{6}$$

$$= \mathbf{Rel}_T(\mathcal{H}|s_{1:t-1}),$$

which is precisely the RHS of Equation (3). □

**Lemma B.5** (Replacing Supremum by Expectation). *For any $V^{(t)} \in \mathcal{X}^{K(T-t)}$, there exists a set of $K$ variables $Z_t = \{z_{t,k}\}_{k \in [K]}$, such that*

$$\sup_{\mathcal{D}_t^{\mathcal{X}} \in \Delta_\sigma(\mathcal{X})} \mathop{\mathbb{E}}_{x_t \sim \mathcal{D}_t^{\mathcal{X}}} \left[ \mathfrak{R}(-L^{\mathrm{r}}(\cdot, s_{1:t-1}), V^{(t)} \cup \{x_t\}) \right] \leq \mathop{\mathbb{E}}_{Z_t \overset{\mathrm{iid}}{\sim} \mathcal{U}(\mathcal{X})} \left[ \mathfrak{R}(-L^{\mathrm{r}}(\cdot, s_{1:t-1}), V^{(t)} \cup Z_t) \right] + \beta.$$

*Proof.* To establish the monotonicity property, we need to show that the random instance $x_t$ drawn from a smooth distribution belongs to a set of uniform i.i.d. hints with high probability. This is where the coupling lemma comes in. For the smooth distribution $\mathcal{D}_t \in \Delta_\sigma(\mathcal{X})$ that achieves the supremum (assume the supremum is achievable), Lemma B.1 shows the existence of a coupling $\Pi$ on $(x_t, z_{t,1}, \cdots, z_{t,K})$ such that $x_t$ is distributed according to $\mathcal{D}_t^{\mathcal{X}}$ and $Z_t = \{z_{t,k}\}_{k \in [K]}$ are uniformly and independently distributed. We thus have

$$\sup_{\mathcal{D}_t \in \Delta_\sigma(\mathcal{X})} \mathop{\mathbb{E}}_{x_t \sim \mathcal{D}_t} \left[ \mathfrak{R}(-L^{\mathrm{r}}(\cdot, s_{1:t-1}), V^{(t)} \cup \{x_t\}) \right] = \mathop{\mathbb{E}}_{\Pi} \left[ \mathfrak{R}(-L^{\mathrm{r}}(\cdot, s_{1:t-1}), V^{(t)} \cup \{x_t\}) \right]. \tag{7}$$

This joint distribution $\Pi$ has the property that event the $E_t \overset{\mathrm{def}}{=} \{x_t \in Z_t\}$ happens with high probability. We now upper bound the expected value by conditioning on $E_t$ and $\bar{E}_t$ respectively.

Conditioned on $E_t$, we apply the monotonicity of regularized Rademacher complexity (Lemma 4.1) recursively and obtain

$$\mathfrak{R}(-L^{\mathrm{r}}(\cdot, s_{1:t-1}), V^{(t)} \cup \{x_t\}) \leq \mathfrak{R}(-L^{\mathrm{r}}(\cdot, s_{1:t-1}), V^{(t)} \cup Z_t). \tag{8}$$

Conditioned on $\bar{E}_t$, we skirt the monotonicity issue by directly using upper and lower bounds on the regularized Rademacher complexity. To be more precise, we use Lemma B.6 in Appendix B.6 to show that

$$\mathfrak{R}(-L^{\mathrm{r}}(\cdot, s_{1:t-1}), V^{(t)} \cup \{x_t\}) \leq TK \leq TK + \left(\mathfrak{R}(-L^{\mathrm{r}}(\cdot, s_{1:t-1}), V^{(t)} \cup Z_t) + T\right)$$

$$\leq \mathfrak{R}(-L^{\mathrm{r}}(\cdot, s_{1:t-1}), V^{(t)} \cup Z_t) + 2TK. \tag{9}$$

Finally, we expand the right hand side of Equation (7) by conditioning on $E_t$ and $\bar{E}_t$ respectively. Putting Equations (8) and (9) together, we obtain

$$
\begin{aligned}
(7) = {}& \Pr[E_t] \cdot \mathop{\mathbb{E}}_{\Pi}\left[\mathfrak{R}(-L^{\mathrm{r}}(\cdot, s_{1:t-1}), V^{(t)} \cup \{x_t\})\middle| E_t\right] \\
&+ \Pr[\bar{E}_t] \cdot \mathop{\mathbb{E}}_{\Pi}\left[\mathfrak{R}(-L^{\mathrm{r}}(\cdot, s_{1:t-1}), V^{(t)} \cup \{x_t\})\middle| \bar{E}_t\right] \\
\leq {}& \Pr[E_t] \cdot \mathop{\mathbb{E}}_{\Pi}\left[\mathfrak{R}(-L^{\mathrm{r}}(\cdot, s_{1:t-1}), V^{(t)} \cup Z_t)\middle| E_t\right] \\
&+ \Pr[\bar{E}_t] \cdot \mathop{\mathbb{E}}_{\Pi}\left[\mathfrak{R}(-L^{\mathrm{r}}(\cdot, s_{1:t-1}), V^{(t)} \cup Z_t) + 2TK\middle| \bar{E}_t\right] \\
= {}& \mathop{\mathbb{E}}_{\Pi}\left[\mathfrak{R}(-L^{\mathrm{r}}(\cdot, s_{1:t-1}), V^{(t)} \cup Z_t)\right] + \Pr[\bar{E}_t] \cdot 2TK.
\end{aligned}
$$

Since $\Pi$ has uniform marginal distribution on $Z_t$, and that $\Pr[\bar{E}_t] \cdot 2TK \leq (1-\sigma)^K \cdot 2TK \leq \beta$, we further obtain

$$(7) \leq \mathop{\mathbb{E}}_{Z_t \overset{\mathrm{iid}}{\sim} \mathcal{U}(\mathcal{X})}\left[\mathfrak{R}(-L^{\mathrm{r}}(\cdot, s_{1:t-1}), V^{(t)} \cup Z_t)\right] + \beta,$$

thus completes the proof. $\qquad\square$

## B.6 Upper and Lower Bounds on the Relaxation

**Lemma B.6** (Upper and Lower Bounds on the Relaxation). *For all $t \in [T]$, all sequence $s_{1:T}$, and all instance set $Z$ of size no larger than $(T-t)K$,*

$$-\frac{T}{2} \leq \mathfrak{R}(-L^{\mathrm{r}}(\cdot, s_{1:t}), Z) \leq TK.$$

*Proof.* Let $I$ be the number of instances in set $Z$ and let $\mathcal{E} = (\epsilon_1, \cdots, \epsilon_I)$ be the random labels associated with them. By convexity of the supremum,

$$
\begin{aligned}
\mathfrak{R}(-L_t, Z) &= \mathop{\mathbb{E}}_{\mathcal{E} \overset{\mathrm{iid}}{\sim} \mathcal{U}(\mathcal{Y})}\left[\sup_{h \in \mathcal{H}}\left\{\sum_{i=1}^{I} \epsilon_i h(z_i) - L^{\mathrm{r}}(h, s_{1:t})\right\}\right] \geq \sup_{h \in \mathcal{H}}\left\{\mathop{\mathbb{E}}_{\mathcal{E} \overset{\mathrm{iid}}{\sim} \mathcal{U}(\mathcal{Y})}\left[\sum_{i=1}^{I} \epsilon_i h(z_i) - L^{\mathrm{r}}(h, s_{1:t})\right]\right\} \\
&= \sup_{h \in \mathcal{H}}\sum_{i=1}^{t} \underbrace{(-l^{\mathrm{r}}(h(x_t), y_t))}_{\geq -1} \geq -T.
\end{aligned}
$$

For the upper bound, we notice that $\forall \mathcal{E}, h$,

$$\sum_{i=1}^{I} \epsilon_i h(z_i) - L^{\mathrm{r}}(h, s_{1:t}) \leq I + t \leq (T-t)K + t \leq TK.$$

So the $\mathfrak{R}(-L^{\mathrm{r}}(\cdot, s_{1:t}), Z)$ also has an upper bound of $TK$. $\qquad\square$

## B.7 Remark on the Requirement of Fresh Dataset

In order to beat the adaptive adversary, the learner needs to sample fresh random hints in each round. Otherwise, the adversary can enforce high regret by correlating future labels with the history. More precisely, we will see that the *matching randomness* argument in Inequality (4) uses the crucial fact that $V^{(t)}$ is a *fresh* dataset that is uniformly distributed independent of the interactions in the past. If $V^{(t)}$ is reused, then the adaptive adversary has the power to correlate $s_{1:t-1}$ with $V^{(t)}$ such that $V^{(t)}$ is no longer unbiased conditioned on the history. In this case, the algorithm fails to mimic the randomization in the relaxation, and the matching-randomness argument breaks down.

Another important property of the fresh self-generated hints $v_{i,k}^{(t)}$ is that they are identically distributed with the real hints $z_{i,k}$ in the coupling. Nevertheless, the analysis has to unite the fact that the learner can only access $v_{i,k}^{(t)}$s, and the monotonicity property (lemma B.5) is based on $z_{i,k}$. This point is subtle because it is impossible for the self-generated hints to really tell the future (i.e., ensure $x_t \in \{v_{t,k}^{(t-1)}\}_{k \in [K]}$), since they are not controlled by the coupling $\Pi$. This issue is taken care of by Equation (6). We can see that it is sufficient for the uncoupled hints in $V^{(t-1)}$ to resemble the coupled hints $Z_t$ at *distribution* level. This distributional resemblence is not achievable if $V^{(t-1)}$ were not independent with the past.

# C Oracle-Efficient Online Binary Classification

## C.1 Information Theoretic Lemmas

For two probability distributions $P$ and $Q$ over the same domain $\mathcal{X}$, let $\chi^2(P,Q) = \sum_{x \in \mathcal{X}} P(x)^2/Q(x) - 1$ be the $\chi^2$-divergence. The following lemma upper bounds the TV distance by the $\chi^2$-divergence; a proof could be found in [Tsy09, Chapter 2].

**Lemma C.1** (From TV to $\chi^2$). *The following relations hold:*

$$\mathrm{TV}(P,Q) \leq \sqrt{\frac{1}{2}\log(1 + \chi^2(Q,P))} \leq \sqrt{\frac{\chi^2(Q,P)}{2}}.$$

The following statement is the well-known Ingster's $\chi^2$ method, and we refer to the excellent book [IS03] for a general treatment.

**Lemma C.2** (Ingster's $\chi^2$ method). *For a mixture distribution $\mathbb{E}_{\theta \sim \pi}[Q_\theta]$ and a generic distribution $P$, the following identity holds:*

$$\chi^2\left(\mathop{\mathbb{E}}_{\theta \sim \pi}[Q_\theta], P\right) = \mathop{\mathbb{E}}_{\theta,\theta' \sim \pi}\left[\mathop{\mathbb{E}}_{x \sim P}\left(\frac{Q_\theta(x)Q_{\theta'}(x)}{P(x)^2}\right)\right] - 1,$$

*where $\theta'$ is an independent copy of $\theta$.*

## C.2 Proof of Theorem 3.2

**Theorem 3.2** (Restated). *In the setting of binary classification with $\sigma$-smoothed adversaries, Algorithm 2 has regret that is at most $\widetilde{O}\left(\min\left\{\sqrt{dT\sigma^{-1/2}}, \sqrt{T(d|\mathcal{X}|)^{1/2}}\right\}\right)$. Furthermore, Algorithm 2 is a* proper *learning oracle-efficient algorithm: at every round t, this algorithm uses a single ERM oracle call on a history that is of length $t + O(T/\sqrt{\sigma})$ with high probability.*

In the remainder of this section, we present a proof of the regret upper bound $\widetilde{O}(\sqrt{dT\sigma^{-1/2}})$ in Theorem 3.2 when $\sigma \geq d/|\mathcal{X}|$. The proof of the other case $\sigma < d/|\mathcal{X}|$ is slightly different and will be presented in Appendix C.6.

To prove the upper bound, we will use the relaxation framework for analyzing the stability of FTPL. Our proof establishes the connection between the relaxation and FTPL frameworks, which we believe will be useful for the design of online algorithms more generally.

Writing $s = (x, y)$ and $L(h, s) = l(h(x), y) = -yh(x)/2$, the relaxation is defined as

$$\mathbf{Rel}_T(\mathcal{H} \mid s_{1:t}) = \mathop{\mathbb{E}}_{R^{(t+1)}} \left[ \sup_{h \in \mathcal{H}} \left( - \sum_{i=1}^{N^{(t+1)}} L(h, \widetilde{s}_i^{(t+1)}) - \sum_{\tau=1}^{t} L(h, s_\tau) \right) \right] + \eta(T - t), \quad (10)$$

where

$$\eta = \frac{1}{\sqrt{n\sigma}} + c\sqrt{\frac{d \log T}{n\sigma}} + \frac{n\sigma}{4T^2 \log T} + e^{-n/8} \in \widetilde{O}\left( \sqrt{\frac{d}{n\sigma}} \right),$$

with an absolute constant $c > 0$ given in Lemma C.5 later, and $R^{(t)} = (N^{(t)}, \{\widetilde{s}_i\}_{i \in N^{(t)}})$ is the fresh randomness generated at the beginning of time $t$, which is independent of $\{s_\tau\}_{\tau < t}$ generated by the adversary. The relaxation here is similar to Equation (2), where the key difference is a different generation process for the hint set and an additional term $\eta(T - t)$ to account for the stability.

Let $\mathcal{Q}_t$ be the distribution of the learner's action $h_t \in \mathcal{H}$ in Algorithm 2, then the relaxation in Equation (10) is admissible with respect to Algorithm 2 if the following two conditions hold:

$$\sup_{\mathcal{D}_t \in \Delta_\sigma(\mathcal{X})} \mathop{\mathbb{E}}_{x_t \sim \mathcal{D}_t} \sup_{y_t} \left[ \mathop{\mathbb{E}}_{h_t \sim \mathcal{Q}_t} [L(h_t, s_t)] + \mathbf{Rel}_T(\mathcal{H} \mid s_{1:t}) \right] \leq \mathbf{Rel}_T(\mathcal{H} \mid s_{1:t-1}), \quad \forall s_{1:t-1} \quad (11)$$

$$\mathbf{Rel}_T(\mathcal{H} \mid s_{1:T}) \geq - \inf_{h \in \mathcal{H}} L(h, s_{1:T}). \quad (12)$$

According to proposition 4.1, if both Inequalities (11) and (12) hold, the expected regret of Algorithm 2 will satisfy

$$\mathbb{E}[\text{Regret}(T)] \leq \mathbf{Rel}_T(\mathcal{H} \mid \emptyset) + O(\sqrt{T}) = \mathop{\mathbb{E}}_{R^{(1)}} \left[ \sup_{h \in \mathcal{H}} \left( - \sum_{i=1}^{N^{(1)}} L(h, \widetilde{s}_i^{(1)}) \right) \right] + \eta T + O(\sqrt{T})$$

$$\overset{(a)}{=} O\left( \mathop{\mathbb{E}}_{N^{(1)}} \left[ \sqrt{dN^{(1)}} \right] + \eta T + \sqrt{T} \right) \overset{(b)}{=} O\left( \sqrt{dn} + \eta T + \sqrt{T} \right),$$

and Theorem 3.2 follows from the choices $n = T/\sqrt{\sigma}$ and $\eta = O(\sqrt{d/n\sigma})$. In the above inequality, step (a) follows from random labels and the upper bound $O(\sqrt{nd})$ on the Rademacher complexity of $\mathcal{H}$ over $n$ points, and step (b) is due to Jensen's inequality and $\mathbb{E}[N^{(1)}] = n$.

Now it remains to verify Inequalities (11) and (12). It is not hard to verify Inequality (12): this follows from the fact that for any random variable $\lambda$, $\mathbb{E}[\sup_\lambda X_\lambda] \geq \sup_\lambda \mathbb{E}[X_\lambda]$ and $\mathbb{E}_R[L(h, \widetilde{s}_i)] = 0$. The key technical difficulty is in the proof of Inequality (11). To overcome this challenge, we will show that the *stability* of learner's distribution $\mathcal{Q}_t$ implies the admissibility of the relaxation, where the stability is measured via

$$\text{Stability} = \mathop{\mathbb{E}}_{s_t \sim \mathcal{D}_t} \left( \mathop{\mathbb{E}}_{h_t \sim \mathcal{Q}_t} [L(h_t, s_t)] - \mathop{\mathbb{E}}_{h_{t+1} \sim \mathcal{Q}_{t+1}} [L(h_{t+1}, s_t)] \right).$$

Note that here $s_t \sim \mathcal{D}_t$ denotes both the instance and its label and $\mathcal{D}_t$'s marginal over $\mathcal{X}$ is $\sigma$-smooth.

The following lemma formalizes the discussion in Section 4.2 and shows that a small TV distance and modified generalization error suffice to ensure the stability of the algorithm, which in turn implies the admissibility of the relaxation. This result could be of independent interest. The proof can be found in Appendix C.3.

**Lemma C.3** (TV + Generalization ⇒ Stability ⇒ Admissibility)**.** *Let $\mathcal{Q}_t$ denote learner's distribution over $\mathcal{H}$ in Algorithm 2 at round $t$, $\mathcal{D}_t$ be adversary's distribution at time $t$ (given the history $s_1, \cdots, s_{t-1}$), $s_t \sim \mathcal{D}_t$ be the realized adversarial instance at time $t$, and $s'_t$ be an independent copy $s'_t \sim \mathcal{D}_t$. It holds that*

$$\mathop{\mathbb{E}}_{s_t \sim \mathcal{D}_t} \left( \mathop{\mathbb{E}}_{h_t \sim \mathcal{Q}_t} [L(h_t, s_t)] + \mathbf{Rel}_T(\mathcal{H} \mid s_{1:t}) \right) - \mathbf{Rel}_T(\mathcal{H} \mid s_{1:t-1})$$

$$\leq \mathop{\mathbb{E}}_{s_t \sim \mathcal{D}_t} \left( \mathop{\mathbb{E}}_{h_t \sim \mathcal{Q}_t} [L(h_t, s_t)] - \mathop{\mathbb{E}}_{h_{t+1} \sim \mathcal{Q}_{t+1}} [L(h_{t+1}, s_t)] \right) - \eta$$

$$\leq \text{TV}(\mathcal{Q}_t, \mathop{\mathbb{E}}_{s_t \sim \mathcal{D}_t} [\mathcal{Q}_{t+1}]) + \underbrace{\mathop{\mathbb{E}}_{s_t, s'_t \sim \mathcal{D}_t; R^{(t+1)}} [L(h_{t+1}, s'_t) - L(h_{t+1}, s_t)]}_{\textit{Modified Generalization Error}} - \eta.$$

Lemma C.3 shows that, in order to prove the admissibility of the relaxation in Equation (10), it remains to upper bound the TV distance and the generalization error, respectively.

Our next lemma provides an upper bound on the TV distance between $\mathcal{Q}_t$ and the mixture distribution $\mathbb{E}_{s_t \sim \mathcal{D}_t}[\mathcal{Q}_{t+1}]$.

**Lemma C.4** (Upper Bound of TV Distance). *Let $\mathcal{Q}^t$ be the distribution over $h_t$ at time $t$. We have*

$$\sup_{\mathcal{D}_t \in \Delta_\sigma(\mathcal{S})} \mathrm{TV}\left(\mathcal{Q}_t, \; \mathbb{E}_{s_t \sim \mathcal{D}_t}[\mathcal{Q}_{t+1}]\right) \leq \frac{1}{\sqrt{n\sigma}}.$$

The key ingredient in the proof of Lemma C.4 is the Poissonization, which ensures the independence of the number of $+1$ (or $-1$) labels across instances, and enables us to write down the mixture distribution of inputs to the ERM oracle in a compact form. The proof of Lemma C.4 is presented in Appendix C.4.

The following lemma upper bounds the modified generalization error for any smooth distribution $\mathcal{D}_t$. The proof of this lemma is based on the discussions in Section 4.4, and we formally present it in Appendix C.5.

**Lemma C.5** (Upper Bound of Generalization Error). *Under the notations of Lemma C.3, it holds for an absolute constant $c > 0$ (independent of $(n, d, T, \sigma)$) that*

$$\sup_{\mathcal{D}_t \in \Delta_\sigma(\mathcal{X})} \left\{ \mathbb{E}_{s_t, s'_t \sim \mathcal{D}_t; R^{(t+1)}} \left[ L(h_{t+1}, s'_t) - L(h_{t+1}, s_t) \right] \right\} \leq c\sqrt{\frac{d \log T}{n\sigma}} + \frac{n\sigma}{4T^2 \log T} + e^{-n/8}.$$

Now the claimed result of Theorem 3.2 when $\sigma \geq d/|\mathcal{X}|$ follows from Lemma C.3, Lemma C.4, and Lemma C.5.

## C.3  Proof of Lemma C.3

To prove this lemma, we first introduce some notations. For $t \in [T] \cup \{0\}$, let $r^t \in \mathbb{Z}^{\mathcal{X}}$ be the $|\mathcal{X}|$-dimensional random vector with $r^t(x)$ defined to be the difference between the number of $+1$ and $-1$ labels in the self-generated samples and the history up to time $t$ on instance $x$. Formally,

$$r^t(x) = \sum_{i=1}^{N^{(t+1)}} \widetilde{y}_i^{(t+1)} \cdot \mathbf{1}(\widetilde{x}_i^{(t+1)} = x) + \sum_{\tau=1}^{t} y_\tau \cdot \mathbf{1}(x_\tau = x).$$

Let $\mathcal{P}^t$ be the distribution of $r^t$. Also recall that $R^{(t)} = (N^{(t)}, \{\widetilde{s}_i\}_{i \in N^{(t)}})$ is the fresh randomness generated at the beginning of time $t$.

Using the definitions of $r^t$, $\mathcal{P}^t$ and $R^{(t)}$, the following chain of inequalities holds for any fixed $s_t$:

$\mathbb{E}_{h_t \sim \mathcal{Q}_t}[L(h_t, s_t)] + \mathbf{Rel}_T(\mathcal{H} \mid s_{1:t})$

$\overset{(a)}{=} \mathbb{E}_{\mathcal{P}^{t-1}}[L(\mathsf{opt}(r^{t-1}), s_t)] - \mathbb{E}_{R^{(t+1)}} \left[ \sum_{i=1}^{N^{(t+1)}} L(\mathsf{opt}(r^t), \widetilde{s}_i^{(t+1)}) + \sum_{\tau=1}^{t} L(\mathsf{opt}(r^t), s_\tau) \right] + \eta(T - t)$

$= \mathbb{E}_{\mathcal{P}^{t-1}}[L(\mathsf{opt}(r^{t-1}), s_t)] - \mathbb{E}_{\mathcal{P}^t}[L(\mathsf{opt}(r^t), s_t)] + \eta(T - t)$

$\quad - \mathbb{E}_{R^{(t+1)}} \left[ \sum_{i=1}^{N^{(t+1)}} L(\mathsf{opt}(r^t), \widetilde{s}_i^{(t+1)}) + \sum_{\tau=1}^{t-1} L(\mathsf{opt}(r^t), s_\tau) \right]$

$\leq \mathbb{E}_{h_t \sim \mathcal{Q}_t}[L(h_t, s_t)] - \mathbb{E}_{h_{t+1} \sim \mathcal{Q}_{t+1}}[L(h_{t+1}, s_t)] + \eta(T - t) + \mathbb{E}_{R^{(t+1)}} \left[ \sup_{h \in \mathcal{H}} \left( -\sum_{i=1}^{N^{(t+1)}} L(h, \widetilde{s}_i^{(t+1)}) - \sum_{\tau=1}^{t-1} L(h, s_\tau) \right) \right]$

$\overset{(b)}{=} \mathbb{E}_{h_t \sim \mathcal{Q}_t}[L(h_t, s_t)] - \mathbb{E}_{h_{t+1} \sim \mathcal{Q}_{t+1}}[L(h_{t+1}, s_t)] + \eta(T - t) + \mathbb{E}_{R^{(t)}} \left[ \sup_{h \in \mathcal{H}} \left( -\sum_{i=1}^{N^{(t)}} L(h, \widetilde{s}_i^{(t)}) - \sum_{\tau=1}^{t-1} L(h, s_\tau) \right) \right]$

$= \mathbb{E}_{h_t \sim \mathcal{Q}_t}[L(h_t, s_t)] - \mathbb{E}_{h_{t+1} \sim \mathcal{Q}_{t+1}}[L(h_{t+1}, s_t)] - \eta + \mathbf{Rel}_T(\mathcal{H} \mid s_{1:t-1}),$

where (a) uses the definition of $\mathrm{opt}(r^t)$, and (b) is due to the fact that $R^{(t+1)}$ is an independent copy of $R^{(t)}$ conditioned on $\{s_\tau\}_{\tau<t}$. This implies the first inequality of Lemma C.3.

For the second inequality, we further take the expectation with respect to $s_t \sim \mathcal{D}_t$, and note that $\mathcal{Q}_t$ and $\mathbf{Rel}_T(\mathcal{H} \mid s_{1:t-1})$ are independent of $s_t$, while $\mathcal{Q}_{t+1}$ depends on $s_t$:

$$
\mathop{\mathbb{E}}_{s_t \sim \mathcal{D}_t} \left( \mathop{\mathbb{E}}_{h_t \sim \mathcal{Q}_t} [L(h_t, s_t)] + \mathbf{Rel}_T(\mathcal{H} \mid s_{1:t}) \right) - \mathbf{Rel}_T(\mathcal{H} \mid s_{1:t-1})
$$

$$
\leq \mathop{\mathbb{E}}_{s_t \sim \mathcal{D}_t} \mathop{\mathbb{E}}_{h_t \sim \mathcal{Q}_t} [L(h_t, s_t)] - \mathop{\mathbb{E}}_{s_t \sim \mathcal{D}_t} \mathop{\mathbb{E}}_{h_{t+1} \sim \mathcal{Q}_{t+1}} [L(h_{t+1}, s_t)] - \eta
$$

$$
\leq \mathop{\mathbb{E}}_{s_t \sim \mathcal{D}_t} \mathop{\mathbb{E}}_{h_t \sim \mathcal{Q}_t} [L(h_t, s_t)] - \mathop{\mathbb{E}}_{s_t, s_t' \sim \mathcal{D}_t} \mathop{\mathbb{E}}_{h_{t+1} \sim \mathcal{Q}_{t+1}} [L(h_{t+1}, s_t')]
$$

$$
+ \mathop{\mathbb{E}}_{s_t, s_t' \sim \mathcal{D}_t} \mathop{\mathbb{E}}_{h_{t+1} \sim \mathcal{Q}_{t+1}} [L(h_{t+1}, s_t')] - \mathop{\mathbb{E}}_{s_t \sim \mathcal{D}_t} \mathop{\mathbb{E}}_{h_{t+1} \sim \mathcal{Q}_{t+1}} [L(h_{t+1}, s_t)] - \eta
$$

$$
\overset{(c)}{=} \mathop{\mathbb{E}}_{s_t' \sim \mathcal{D}_t} \mathop{\mathbb{E}}_{h_t \sim \mathcal{Q}_t} [L(h_t, s_t')] - \mathop{\mathbb{E}}_{s_t' \sim \mathcal{D}_t} \mathop{\mathbb{E}}_{h_{t+1} \sim \mathbb{E}_{s_t \sim \mathcal{D}_t}[\mathcal{Q}_{t+1}]} [L(h_{t+1}, s_t')]
$$

$$
+ \mathop{\mathbb{E}}_{s_t, s_t' \sim \mathcal{D}_t} \mathop{\mathbb{E}}_{h_{t+1} \sim \mathcal{Q}_{t+1}} [L(h_{t+1}, s_t')] - \mathop{\mathbb{E}}_{s_t \sim \mathcal{D}_t} \mathop{\mathbb{E}}_{h_{t+1} \sim \mathcal{Q}_{t+1}} [L(h_{t+1}, s_t)] - \eta
$$

$$
\overset{(d)}{\leq} \mathrm{TV}(\mathcal{Q}_t, \mathop{\mathbb{E}}_{s_t \sim \mathcal{D}_t}[\mathcal{Q}_{t+1}]) + \mathop{\mathbb{E}}_{s_t, s_t' \sim \mathcal{D}_t; R^{(t+1)}} \left[ L(h_{t+1}, s_t') - L(h_{t+1}, s_t) \right] - \eta,
$$

where (c) follows from the independence of $h_t \sim \mathcal{Q}_t$ and $(s_t, s_t')$, and (d) is due to $| \mathbb{E}_{X \sim P}[f(X)] - \mathbb{E}_{X \sim Q}[f(X)] | \leq \mathrm{TV}(P, Q)$ for every measurable function $f$ with $\|f\|_\infty \leq 1$.

## C.4  Upper Bounding TV Distance: Proof of Lemma C.4

Using the notations in lemma C.3, since $h_t$ in Algorithm 2 only depends on the vector $r^{t-1}$, the ERM objective could be written as a quantity depending only on $r^{t-1}$ and $h \in \mathcal{H}$. We write $h_t = \mathrm{opt}_{\mathcal{H}, l}(r^{t-1})$ in the sequel, and then $\mathrm{opt}_{\mathcal{H}, l}(r^{t-1}) \sim \mathcal{Q}_t$ as $r^{t-1} \sim \mathcal{P}^{t-1}$. Therefore, the data-processing inequality shows that

$$
\mathrm{TV}(\mathcal{Q}_t, \mathop{\mathbb{E}}_{s_t \sim \mathcal{D}_t}[\mathcal{Q}_{t+1}]) \leq \mathrm{TV}(\mathcal{P}^{t-1}, \mathop{\mathbb{E}}_{s_t \sim \mathcal{D}_t}[\mathcal{P}^t]),
$$

and is suffices to upper bound the TV distance $\mathrm{TV}(\mathcal{P}^{t-1}, \mathbb{E}_{s_t \sim \mathcal{D}_t}[\mathcal{P}^t])$.

Let us first create a better understanding of the structures of the distributions $\mathcal{P}^{t-1}$ and $\mathcal{P}^t$. Without loss of generality we assume that $\mathcal{X}$ is discrete (the case of continuous $\mathcal{X}$ can be dealt by analyzing the appropriate Poisson point process). Let $n_+(x), n_-(x)$ be the numbers of $+1$ and $-1$ labels, respectively, given instance $x$ in the self-generated samples:

$$
n_+(x) = \sum_{i=1}^{N} \mathbf{1}(\widetilde{x}_i = x, \widetilde{y}_i = +1) \quad \text{and} \quad n_-(x) = \sum_{i=1}^{N} \mathbf{1}(\widetilde{x}_i = x, \widetilde{y}_i = -1).
$$

As each $\widetilde{x}_i$ is uniformly distributed on $\mathcal{X}$ and $\widetilde{y}_i \sim \mathcal{U}(\{\pm 1\})$, by the subsampling property of the Poisson distribution, the $2|\mathcal{X}|$ random variables $\{n_\pm(x)\}_{x \in \mathcal{X}}$ are i.i.d. distributed as $\mathrm{Poi}(n/2|\mathcal{X}|)$. This independence implied by the Poisson distribution plays a key role in the analysis. Moreover, $r^0(x) = n_+(x) - n_-(x)$, so $\mathcal{P}^0$ is determined by the joint distribution of $\{n_\pm(x)\}_{x \in \mathcal{X}}$.

As we move to general $t$, note that the only contribution of the historic data $\{s_\tau\}_{\tau < t}$ to both $\mathcal{P}^{t-1}$ and $\mathcal{P}^t$ is a common translation independent of $\mathcal{P}^0$. Since the TV distance is translation invariant, it suffices to upper bound $\mathrm{TV}(\mathcal{P}^0, \mathbb{E}_{s_1}[\mathcal{P}^1])$. Let $n_\pm^1(x) = n_\pm(x) + \mathbf{1}(x_1 = x, y_1 = \pm 1)$, it holds that $r^1(x) = n_+^1(x) - n_-^1(x)$. Consequently, let $P$ and $Q$ be the probability distributions of $\{n_\pm(x)\}_{x \in \mathcal{X}}$ and $\{n_\pm^1(x)\}_{x \in \mathcal{X}}$, respectively, the data-processing inequality implies that $\mathrm{TV}(\mathcal{P}^0, \mathbb{E}_{s_1}[\mathcal{P}^1]) \leq \mathrm{TV}(P, Q)$.

As discussed above, the distribution $P$ is a product Poisson distribution:

$$
P(\{n_\pm(x)\}) = \prod_{x \in \mathcal{X}} \prod_{y \in \{\pm\}} \mathbb{P}(\mathrm{Poi}(n/2|\mathcal{X}|) = n_y(x)).
$$

As for the distribution $Q$, it could be obtained from $P$ in the following way: the smooth adversary draws $x^\star \sim \mathcal{D}$, independent of $\{n_\pm(x)\}_{x \in \mathcal{X}} \sim P$, for some $\sigma$-smooth distribution $\mathcal{D} \in \Delta_\sigma(\mathcal{X})$. He then chooses a label $y^\star = y(x^\star) \in \{\pm 1\}$ as a function of $x^\star$, and sets

$$n^1_{y(x^\star)}(x^\star) = n_{y(x^\star)}(x^\star) + 1, \qquad \text{and} \qquad n^1_y(x) = n_y(x), \quad \forall (x,y) \neq (x^\star, y(x^\star)).$$

Consequently, given a $\sigma$-smooth distribution $\mathcal{D}$ and a labeling function $y : \mathcal{X} \to \{\pm\}$ used by the adversary, the distribution $Q$ is a mixture distribution $Q = \mathbb{E}_{x^\star \sim \mathcal{D}^\mathcal{X}}[Q_{x^\star}]$, with

$$Q_{x^\star}(\{n_\pm(x)\}) = \mathbb{P}(\text{Poi}(n/2|\mathcal{X}|) = n_{y(x^\star)}(x^\star) - 1) \times \prod_{(x,y) \neq (x^\star, y(x^\star))} \mathbb{P}(\text{Poi}(n/2|\mathcal{X}|) = n_y(x)).$$

To upper bound the TV distance between a mixture distribution $Q$ and a base distribution $P$, we will rely on the smoothness properties of $\mathcal{D}$, in particular, that the probability of collision between two independent draws $x_1^\star, x_2^\star \sim \mathcal{D}$ is small. To formally address this, we make use of two technical lemmas, first to upperbound the TV distance in terms of the $\chi^2$ distance, and second to use the Ingster's method for bounding the $\chi^2$ distance between a mixture distribution and a base distribution. See Lemma C.1 and Lemma C.2 in the Appendix C.1 for more details. Let $x_1^\star, x_2^\star$ be an arbitrary pair of instance. Using the closed-form expressions of distributions $P$ and $Q_{x^\star}$, it holds that

$$\frac{Q_{x_1^\star}(\{n_\pm(x)\}) Q_{x_2^\star}(\{n_\pm(x)\})}{P(\{n_\pm(x)\})^2} = \frac{2|\mathcal{X}| n_{y(x_1^\star)}(x_1^\star)}{n} \cdot \frac{2|\mathcal{X}| n_{y(x_2^\star)}(x_2^\star)}{n}.$$

Using the fact that $\{n_\pm(x)\}_{x \in \mathcal{X}}$ are i.i.d. distributed as $\text{Poi}(n/2|\mathcal{X}|)$ under $P$, we have

$$\mathbb{E}_{\{n_\pm(x)\} \sim P} \left( \frac{Q_{x_1^\star}(\{n_\pm(x)\}) Q_{x_2^\star}(\{n_\pm(x)\})}{P(\{n_\pm(x)\})^2} \right) = 1 + \frac{2|\mathcal{X}|}{n} \cdot \mathbf{1}(x_1^\star = x_2^\star).$$

Now using the aforementioned lemmas (Lemma C.1 and Lemma C.2), we have

$$\text{TV}(P, Q) \leq \sqrt{\frac{\chi^2(Q, P)}{2}} = \sqrt{\frac{\chi^2(\mathbb{E}_{x^\star \sim \mathcal{D}}[Q_{x^\star}], P)}{2}} = \sqrt{\frac{|\mathcal{X}|}{n} \cdot \mathbb{E}_{x_1^\star, x_2^\star \sim \mathcal{D}}[\mathbf{1}(x_1^\star = x_2^\star)]}$$

$$= \sqrt{\frac{|\mathcal{X}|}{n} \sum_{x \in \mathcal{X}} \mathcal{D}(x)^2} \overset{(a)}{\leq} \sqrt{\frac{|\mathcal{X}|}{n} \sum_{x \in \mathcal{X}} \mathcal{D}(x) \cdot \frac{1}{\sigma|\mathcal{X}|}} = \frac{1}{\sqrt{\sigma n}},$$

where (a) follows from the definition of a $\sigma$-smooth distribution. This completes the proof.

## C.5   Upper Bounding Generalization Error: Proof of Lemma C.5

In the proof of Lemma C.5, we shall need the following property of smooth distributions which is a slightly strengthened version of the coupling lemma in Lemma B.1. The proof of lemma C.6 is presented in Appendix C.7.

**Lemma C.6.** *Let $X_1, \cdots, X_m \sim Q$ and $P$ be another distribution with a bounded likelihood ratio: $dP/dQ \leq 1/\sigma$. Then using external randomness $R$, there exists an index $I = I(X_1, \cdots, X_m, R) \in [m]$ and a success event $E = E(X_1, \cdots, X_m, R)$ such that $\Pr[E^c] \leq (1 - \sigma)^m$, and*

$$(X_I \mid (E, X_{\backslash I})) \sim P.$$

Fix any realization of the Poissonized sample size $N \sim \text{Poi}(n)$. Choose $m = 4\sigma^{-1} \log T$ in Lemma C.6, and without loss of generality assume that $N$ is an integral multiple of $m$. Since for any $\sigma$-smooth $\mathcal{D}_t$, it holds that

$$\frac{\mathcal{D}_t(s)}{\mathcal{U}(\mathcal{X} \times \{\pm 1\})(s)} = \frac{\mathcal{D}_t(x)}{\mathcal{U}(\mathcal{X})(x)} \cdot \frac{\mathcal{D}_t(y \mid x)}{\mathcal{U}(\{\pm 1\})(y)} \leq \frac{2}{\sigma},$$

the premise of Lemma C.6 holds with parameter $\sigma/2$ for $P = \mathcal{D}_t, Q = \mathcal{U}(\mathcal{X} \times \{\pm 1\})$. Consequently, dividing the self-generated samples $\widetilde{s}_1, \cdots, \widetilde{s}_N$ into $N/m$ groups each of size $m$, and running the procedure in Lemma C.6, we arrive at $N/m$ independent events $E_1, \cdots, E_{N/m}$, each with probability

at least $1 - (1 - \sigma/2)^m \geq 1 - T^{-2}$. Moreover, conditioned on each $E_j$, we can pick an element $u_j \in \{\widetilde{s}_{(j-1)m+1}, \cdots, \widetilde{s}_{jm}\}$ such that

$$(u_j \mid (E_j, \{\widetilde{s}_{(j-1)m+1}, \cdots, \widetilde{s}_{jm}\}\setminus\{u_j\})) \sim \mathcal{D}_t.$$

For notational simplicity we denote the set of unpicked samples $\{\widetilde{s}_{(j-1)m+1}, \cdots, \widetilde{s}_{jm}\}\setminus\{u_j\}$ by $v_j$. As a result, thanks to the mutual independence of different groups and $s_t \sim \mathcal{D}_t$ conditioned on $s_{1:t-1}$ (note that we draw fresh randomness at every round), for $E \triangleq \cap_{j \in [N/m]} E_j$ we have

$$(u_1, \cdots, u_{N/m}, s_t) \mid (E, s_{1:t-1}, v_1, \cdots, v_{N/m}) \overset{\text{iid}}{\sim} \mathcal{D}_t.$$

Consequently, for each $j \in [N/m]$ we have

$$\underset{s_t \sim \mathcal{D}_t, R^{(t+1)}}{\mathbb{E}} [L(h_{t+1}, s_t) \mid E]$$

$$= \underset{s_t \sim \mathcal{D}_t, \widetilde{s}_1, \cdots, \widetilde{s}_N}{\mathbb{E}} \left[ L(\mathsf{opt}(\widetilde{s}_1, \cdots, \widetilde{s}_N, s_{1:t-1}, s_t), s_t) \mid E \right]$$

$$= \underset{v, s_{1:t-1} \mid E}{\mathbb{E}} \left( \underset{s_t, u_1, \cdots, u_{N/m}}{\mathbb{E}} \left[ L(\mathsf{opt}(s_{1:t-1}, v, u_1, \cdots, u_{N/m}, s_t), s_t) \mid (E, s_{1:t-1}, v) \right] \right)$$

$$\overset{(a)}{=} \underset{v, s_{1:t-1} \mid E}{\mathbb{E}} \left( \underset{s_t, u_1, \cdots, u_{N/m}}{\mathbb{E}} \left[ L(\mathsf{opt}(s_{1:t-1}, v, u_1, \cdots, u_{j-1}, s_t, u_{j+1}, \cdots, u_{N/m}, u_j), u_j) \mid (E, s_{1:t-1}, v) \right] \right)$$

$$\overset{(b)}{=} \underset{v, s_{1:t-1} \mid E}{\mathbb{E}} \left( \underset{s_t, u_1, \cdots, u_{N/m}}{\mathbb{E}} \left[ L(\mathsf{opt}(s_{1:t-1}, v, u_1, \cdots, u_{N/m}, s_t), u_j) \mid (E, s_{1:t-1}, v) \right] \right)$$

$$= \underset{s_t \sim \mathcal{D}_t, R^{(t+1)}}{\mathbb{E}} [L(h_{t+1}, u_j) \mid E],$$

where (a) follows from the conditional iid (and thus exchangeable) property of $(u_1, \cdots, u_{N/m}, s_t)$ after the conditioning, and (b) is due to the invariance of the ERM output after any permutation of the inputs. On the other hand, if $s_t', u_1', \cdots, u_{N/m}'$ are independent copies of $s_t \sim \mathcal{D}_t$, by independence it is clear that

$$\underset{s_t, s_t' \sim \mathcal{D}_t, R^{(t+1)}}{\mathbb{E}} [L(h_{t+1}, s_t') \mid E] = \underset{s_t, s_t' \sim \mathcal{D}_t, R^{(t+1)}}{\mathbb{E}} [L(h_{t+1}, u_j') \mid E], \quad \forall j \in [N/m].$$

Consequently, using the shorthand $u_0 = s_t, u_0' = s_t'$, we have

$$\underset{s_t, s_t' \sim \mathcal{D}_t, R^{(t+1)}}{\mathbb{E}} [L(h_{t+1}, s_t') - L(h_{t+1}, s_t) \mid E]$$

$$= \frac{1}{N/m + 1} \underset{s_t, s_t' \sim \mathcal{D}_t, R^{(t+1)}}{\mathbb{E}} \left[ \sum_{j=0}^{N/m} (L(h_{t+1}, u_j') - L(h_{t+1}, u_j)) \,\bigg|\, E \right]$$

$$\leq \frac{1}{N/m + 1} \underset{u_0, \cdots, u_{N/m}, u_0', \cdots, u_{N/m}' \sim \mathcal{D}_t}{\mathbb{E}} \left[ \sup_{h \in \mathcal{H}} \sum_{j=0}^{N/m} (L(h, u_j') - L(h, u_j)) \right]$$

$$\leq \frac{2}{N/m + 1} \underset{u_0, \cdots, u_{N/m} \sim \mathcal{D}_t}{\mathbb{E}} \underset{\epsilon_1 \dots \epsilon_{N/m}}{\mathbb{E}} \left[ \sup_{h \in \mathcal{H}} \sum_{j=0}^{N/m} \epsilon_j h(u_j) \right] \leq c_0 \sqrt{\frac{d}{N/m + 1}},$$

where the last inequality is due to the classical $O(\sqrt{d/n})$ upper bound on the Rademacher complexity, and $c_0 > 0$ in an absolute constant. Note that the union bound gives

$$\Pr[E^c] \leq \sum_{j=1}^{N/m} \Pr[E_j^c] \leq \frac{N}{mT^2},$$

the law of total expectation gives

$$\underset{s_t, s_t' \sim \mathcal{D}_t, R^{(t+1)}}{\mathbb{E}} [L(h_{t+1}, s_t') - L(h_{t+1}, s_t)]$$

$$\leq \underset{s_t, s_t' \sim \mathcal{D}_t, R^{(t+1)}}{\mathbb{E}} [L(h_{t+1}, s_t') - L(h_{t+1}, s_t) \mid E] + \Pr[E^c] \leq c_0 \sqrt{\frac{d}{N/m + 1}} + \frac{N}{mT^2}.$$

Finally, plugging the choice of $m = 4\sigma^{-1} \log T$, taking the expectation of $N \sim \text{Poi}(n)$, and using $\Pr[N > n/2] \geq 1 - e^{-n/8}$ in the above inequality completes the proof of Lemma C.5.

## C.6 Proof of Theorem 3.2 for Small Domain

In this section we complete the proof of the $O(\sqrt{T(d|\mathcal{X}|)^{1/2}})$ upper bound in Theorem 3.2 when $\sigma < d/|\mathcal{X}|$ (and thus $n = T\sqrt{|\mathcal{X}|/d}$). The proof is still through the same relaxation in Equation (10), though we will choose a different parameter $\eta$ and prove a slightly modified version of Lemma C.3:

**Lemma C.7** (Expected TV $\Rightarrow$ Admissibility). *Let $\mathcal{Q}_t$ denote learner's distribution over $\mathcal{H}$ in Algorithm 2 at round $t$, and $s_t \sim \mathcal{D}_t$ be the conditional distribution of $s_t$ given the history $s_1, \cdots, s_{t-1}$. It holds that*

$$\mathop{\mathbb{E}}_{s_t \sim \mathcal{D}_t} \left( \mathop{\mathbb{E}}_{h_t \sim \mathcal{Q}_t} [L(h_t, s_t)] + \textbf{Rel}_T(\mathcal{H} \mid s_{1:t}) \right) - \textbf{Rel}_T(\mathcal{H} \mid s_{1:t-1}) \leq \mathop{\mathbb{E}}_{s_t \sim \mathcal{D}_t} [\text{TV}(\mathcal{Q}_t, \mathcal{Q}_{t+1})] - \eta.$$

*Proof of Lemma C.7.* The analysis is similar to the proof of Lemma C.3. In fact, an intermediate step of Lemma C.3 gives

$$\mathop{\mathbb{E}}_{h_t \sim \mathcal{Q}_t} [L(h_t, s_t)] + \textbf{Rel}_T(\mathcal{H} \mid s_{1:t}) - \textbf{Rel}_T(\mathcal{H} \mid s_{1:t-1}) \leq \mathop{\mathbb{E}}_{h_t \sim \mathcal{Q}_t} [L(h_t, s_t)] - \mathop{\mathbb{E}}_{h_{t+1} \sim \mathcal{Q}_{t+1}} [L(h_{t+1}, s_t)] - \eta.$$

Now using $|\mathbb{E}_{X \sim P}[f(X)] - \mathbb{E}_{X \sim Q}[f(X)]| \leq \text{TV}(P, Q)$ for every measurable function $f$ with $\|f\|_\infty \leq 1$, the RHS is further upper bounded by $\text{TV}(\mathcal{Q}_t, \mathcal{Q}_{t+1}) - \eta$. The proof of Lemma C.7 is completed by taking the expectation over $s_t \sim \mathcal{D}_t$. □

Note that in Lemma C.7, the expectation is outside the TV distance and no smaller than the TV distance when the mixture distribution is inside the expectation compared with Lemma C.3. We can simply upper bound this expected TV distance, with the worst case choice of $s_t$ and apply the data processing inequality, i.e.,

$$\mathop{\mathbb{E}}_{s_t \sim \mathcal{D}_t} [\text{TV}(\mathcal{Q}_t, \mathcal{Q}_{t+1})] \leq \sup_{s_t} \text{TV}(\mathcal{P}^{t-1}, \mathcal{P}^t).$$

Using the similar independence property of Poissonization in Appendix C.4, the target TV distance is at most $\text{TV}(P, Q)$, where $P \sim \text{Poi}(n/2|\mathcal{X}|)$, and $Q$ is a right-translation of $P$ by one. Consequently,

$$\text{TV}(P, Q) \leq \sqrt{\frac{\chi^2(Q, P)}{2}} = \sqrt{\frac{1}{2} \left( \mathop{\mathbb{E}}_{X \sim P} \left[ \left( \frac{X}{n/2|\mathcal{X}|} \right)^2 \right] - 1 \right)} = \sqrt{\frac{|\mathcal{X}|}{n}},$$

so the choice of $\eta = \sqrt{|\mathcal{X}|/n}$ and Lemma C.7 again makes the relaxation in Equation (10) admissible, and we complete the proof of Theorem 3.2.

## C.7 Proof of Lemma C.6

The proof is essentially similar to [BDGR22, Lemma 12], and we include it here for completeness. For each $i \in [m]$, compute the value $p_i = \sigma \frac{dP}{dQ}(X_i)$, which lies in $[0, 1]$ due to the likelihood ratio upper bound. Now we draw an independent Bernoulli random variable $Y_i \sim \text{Bern}(p_i)$, and define the random index $I$ and success event $E$ as follows:

$$E \triangleq \cup_{i=1}^m \{Y_i = 1\},$$
$$I \triangleq \text{a uniformly random element of } \{i \in [m] : Y_i = 1\}.$$

Note that $Y_1, \cdots, Y_m$ are mutually independent, and for each $i \in [m]$,

$$\Pr[Y_i = 1] = \mathop{\mathbb{E}}_{X_i \sim Q}[p_i] = \mathop{\mathbb{E}}_{X_i \sim Q} \left[ \sigma \frac{dP}{dQ}(X_i) \right] = \sigma,$$

we conclude that $\Pr[E] = 1 - (1 - \sigma)^m$. For the second statement, we denote by $r_i$ the external randomness used in drawing $Y_i \sim \text{Bern}(p_i)$, and by $r$ the external randomness used in the definition of $I$. Then for any measurable set $A \subseteq \mathcal{X}$,

$$
\begin{aligned}
&\Pr[X_I \in A \mid E, X_{\backslash I}] \\
&= \sum_{i, r_{\backslash i}, r} \Pr[X_I \in A \mid E, X_{\backslash I}, I = i, r_{\backslash i}, r] \cdot \Pr[I = i, r_{\backslash i}, r \mid E, X_{\backslash I}] \\
&= \sum_{i, r_{\backslash i}, r} \Pr[X_i \in A \mid E, X_{\backslash i}, I = i, r_{\backslash i}, r] \cdot \Pr[I = i, r_{\backslash i}, r \mid E, X_{\backslash I}] \\
&\overset{(a)}{=} \sum_{i, r_{\backslash i}, r} \Pr[X_i \in A \mid Y_i = 1, X_{\backslash i}, r_{\backslash i}, r] \cdot \Pr[I = i, r_{\backslash i}, r \mid E, X_{\backslash I}] \\
&\overset{(b)}{=} \sum_{i, r_{\backslash i}, r} \Pr[X_i \in A \mid Y_i = 1] \cdot \Pr[I = i, r_{\backslash i}, r \mid E, X_{\backslash I}] \\
&\overset{(c)}{=} \sum_{i, r_{\backslash i}, r} P(A) \cdot \Pr[I = i, r_{\backslash i}, r \mid E, X_{\backslash I}] \\
&= P(A),
\end{aligned}
$$

where (a) is due to the event $\{E, I = i, X_{\backslash i}, r_{\backslash i}, r\}$ is the same as $\{Y_i = 1, X_{\backslash i}, r_{\backslash i}, r\}$ as long as the former event $\{E, I = i, X_{\backslash i}, r_{\backslash i}, r\}$ is non-empty (note that empty events do not contribute to the sum), (b) follows from the mutual independence of $(X_i, r_i, Y_i)_{i \in [m]}$ and $r$, (c) is due to

$$
\Pr[X_i \in A \mid Y_i = 1] = \frac{\Pr[X_i \in A, Y_i = 1]}{\Pr[Y_i = 1]} = \frac{1}{\sigma} \mathop{\mathbb{E}}_{X_i \sim Q}\left[ \mathbf{1}(X_i \in A) \sigma \frac{dP}{dQ}(X_i) \right] = P(A).
$$

The above identity shows that the conditional distribution of $X_I$ conditioned on $(E, X_{\backslash I})$ is always $P$, as desired.

## D  Unknown Smoothness Parameters

Suppose we have upper and lower bounds $\sigma_{\max}$ and $\sigma_{\min}$ on the exact value of $\sigma$, i.e., $\sigma_{\min} \leq \sigma \leq \sigma_{\max}$. In this section, we introduce a meta algorithm that uses a geometric doubling approach to incorporate knowledge of $\sigma_{\max}$ and $\sigma_{\min}$ into the algorithms introduced in Section 3.

We start by constructing $\log(\sigma_{\max}/\sigma_{\min})$ experts, where each expert $i$ runs a local version of our algorithm (can be either Algorithm 1 for the real-valued case or Algorithm 2 for the binary case) with parameter $\sigma_i = 2^i \cdot \sigma_{\min}$. We then run Hedge on these experts. Note that the parameter $i^\star$ of the best expert satisfies $\frac{\sigma}{2} \leq \sigma_{i^\star} \leq \sigma$, so the expected regret of this expert matches the expected regret of the same algorithm running on true $\sigma$ up to a constant factor. Therefore, the expected regret of this meta algorithm is comparable to the bound in Theorem 3.1 and 3.2, with an additive term of order at most $O\left(\sqrt{T \log \log(\sigma_{\max}/\sigma_{\min})}\right)$. The number of oracle calls also blows up only by $\log(\sigma_{\max}/\sigma_{\min})$ per round. This could potentially be improved using a more aggressive step size for the Hedge meta algorithm.

## E  Proof of Lower Bounds (Theorem E.1 and Theorem 5.1)

**Theorem E.1** (Limitations of Algorithms). *For any choice of the parameter $n$ in Algorithm 2 (or Algorithm 1), there exists a $\sigma$-smoothed online learning instance such that Algorithm 2 (or Algorithm 1) suffers from at least $\Omega(\min\{T, \sqrt{dT\sigma^{-1/2}}, \sqrt{T(d|\mathcal{X}|)^{1/2}}\})$ expected regret.*

### E.1  Proof of Theorem E.1

This section proves the regret lower bounds for Algorithm 2 and Algorithm 1 stated in Theorem E.1. We split the analysis into two subsections, and in each subsection we prove a large regret both when the sample size parameter $n$ is large and small.

### E.1.1 Lower Bound Analysis for Algorithm 2

We shall only prove the regret lower bound $\Omega(\sqrt{dT}\sigma^{-1/2})$ under the assumption $\sigma \geq \max\{d/|\mathcal{X}|, (d/T)^2\}$, for a smaller $\sigma$ only makes the worst-case regret larger, and the other lower bounds follow from this case by taking $\sigma = d/|\mathcal{X}|$ and $\sigma = (d/T)^2$, respectively. We split the analysis into two cases depending on the choice of parameter $n$.

**Case I: Large $n$.** When $n$ is large, or more specifically, when $n \geq T/\sqrt{\sigma}$, consider the behavior of Algorithm 2 on the following instance. Consider any domain $\mathcal{X}$ where $|\mathcal{X}|$ is an integral multiple of $d$, and partition $\mathcal{X} = \cup_{j=1}^{d} \mathcal{X}_j$ into $d$ sets $\{\mathcal{X}_j\}_{j \in [d]}$ with an equal size. Consider the following hypothesis class:

$$\mathcal{H} = \{h : \mathcal{X} \to \{\pm 1\} \mid h \text{ is a constant on } \mathcal{X}_i, \forall i \in [d]\}.$$

Clearly $\mathcal{H}$ has VC dimension $d$. The adversary chooses a hypothesis $h^\star \in \mathcal{H}$ uniformly at random, and sets $x_t$ to be uniformly distributed on $\mathcal{X}$. As for the label $y_t$, the adversary sets $y_t = h^\star(x_t)$. This adversary is 1-smooth, and the best expert in $\mathcal{H}$ incurs a zero loss under this realizable setting. We claim that for each of the first $\min\{T, c\sqrt{nd}\}$ time steps, for an absolute constant $c > 0$ sufficiently small, Algorithm 2 makes a mistake with $\Omega(1)$ probability. Summing over these steps, the expected regret of Algorithm 2 is then $\Omega(\min\{T, c\sqrt{nd}\})$, which gives Theorem E.1 by our assumption $n \geq T/\sqrt{\sigma}$.

To prove this claim, we need the following lemma.

**Lemma E.2** (Minimum Error on Hallucinated Samples). *For $N \sim \text{Poi}(n)$ hallucinated samples $(x_1, y_1), \cdots, (x_N, y_N)$, if $n \geq d$, it holds that*

$$\mathbb{P}\left(\sum_{i=1}^{N} y_i \cdot \mathbf{1}(x_i \in \mathcal{X}_j) \geq \sqrt{\frac{n}{d}}\right) = \Omega(1), \qquad \forall j \in [d].$$

*Proof.* For $j \in [d]$, let $n_{j,+}, n_{j,-}$ denote the number of hallucinate samples $(x_i, y_i)$ with $x_i \in \mathcal{X}_j$ and $y_i = \pm 1$, respectively. By the Poisson subsampling property, $\{n_{j,\pm}\}_{j \in [d]}$ are mutually independent $\text{Poi}(n/(2d))$ random variables. By definition of $\mathcal{H}$, we have

$$n_{j,+} - n_{j,-} = \sum_{i=1}^{N} y_i \cdot \mathbf{1}(x_i \in \mathcal{X}_j).$$

Consequently, the quantity of interest is $n_{j,+} - n_{j,-}$. As $n/d \geq 1$, by the Poisson tail property, both events $n_{j,+} \geq n/(2d) + \sqrt{n/d}/2$ and $n_{j,-} \leq n/(2d) - \sqrt{n/d}/2$ happen with $\Omega(1)$ probability, and their independence gives the claimed result. $\square$

Since $(d/T)^2 \leq \sigma \leq 1$, we have $T \geq d$ and thus $n \geq T/\sqrt{\sigma} \geq d$, the premise of Lemma E.2 holds. Consequently, at each time step $t \leq \min\{T, c\sqrt{nd}\}$ with $x_t \in \mathcal{X}_j$, with $\Omega(1)$ probability there are at least $\sqrt{n/d}$ net positive labels in the hallucinated samples, while the learner has only observed at most $\alpha c \sqrt{n/d}$ labels in the history with probability at least $1 - 1/\alpha$, by Markov's inequality. By choosing constants $c > 0$ small and $\alpha > 0$ large, the perturbed leader will predict $+1$ depending only on the hallucination, and this prediction is independent of the choice of $h^\star$ and thus incurs an error with probability $1/2$. This proves the claim that before time $\min\{T, c\sqrt{nd}\}$, there is always $\Omega(1)$ probability of error.

**Case II: Small $n$.** Now we turn to the scenario where $n < T/\sqrt{\sigma}$. Consider the following learning instance: choose $\mathcal{X}_0 \subseteq \mathcal{X}$ with $|\mathcal{X}_0| = \sigma|\mathcal{X}| \geq d$, the adversary always chooses $x_t \sim \mathcal{U}(\mathcal{X}_0)$, which is $\sigma$-smooth. Assuming that $|\mathcal{X}_0|$ is an integral multiple of $d$, we partition $\mathcal{X}_0 = \cup_{j=1}^{d} \mathcal{X}_j$ into $d$ subsets with equal size. Condition on each $\mathcal{X}_j$, consider an alternating label sequence:

$$(y_t : x_t \in \mathcal{X}_j)_{t=1}^{T} = (+1, -1, +1, -1, \cdots).$$

The hypothesis class $\mathcal{H}$ consists of $2^d$ functions:

$$\mathcal{H} = \{h : \mathcal{X} \to \{\pm 1\} \mid h \text{ is a constant on } \mathcal{X}_j, \forall j \in [d], \text{ and } h(x) \equiv 1, \forall x \in \mathcal{X}\backslash\mathcal{X}_0\}.$$

Clearly $\mathcal{H}$ has VC dimension $d$, and the best hypothesis in $\mathcal{H}$ incurs a cumulative loss $T/2$.

Now we examine the performance of Algorithm 2. Let $r_j$ be the difference between the number of $+1$ and $-1$ labels in the hallucinated samples with feature in $\mathcal{X}_j$, similar to the proof of Lemma E.2 we have $r_j = n_{j,+} - n_{j,-}$ for independent Poisson random variables $n_{j,+}, n_{j,-} \sim \text{Poi}(n\sigma/2d)$. Suppose that ties are broken by always predicting $-1$ when calling the ERM oracle, we observe that Algorithm 2 always makes a mistake when $x \in \mathcal{X}_j$ and $r_j = 0$ – this is the same counterexample where Follow-The-Leader (FTL) makes a mistake at every step. Moreover, when $r_j \neq 0$, Algorithm 2 makes $T/2$ mistakes, same as the best expert in $\mathcal{H}$. Consequently, the expected regret of Algorithm 2 is at least $T \cdot \mathbb{P}(r_j = 0)$, where

$$\mathbb{P}(r_j = 0) = \mathop{\mathbb{E}}_{N \sim \text{Poi}(n\sigma/d)}\left[\mathbb{P}\left(\text{Bin}(N, \tfrac{1}{2}) = \tfrac{N}{2}\right)\right] = \mathop{\mathbb{E}}_{N \sim \text{Poi}(n\sigma/d)}\left[\Omega\left(\frac{\mathbf{1}(N \text{ is even})}{\sqrt{N+1}}\right)\right]$$

$$\overset{(a)}{=} \Omega\left(\frac{\mathbb{P}_{N \sim \text{Poi}(n\sigma/d)}(N \text{ is even})}{\sqrt{n\sigma/d + 1}}\right) \overset{(b)}{=} \Omega\left(\frac{1}{\sqrt{n\sigma/d + 1}}\right) = \Omega\left(\min\left\{1, \sqrt{\frac{d}{n\sigma}}\right\}\right).$$

In the above display, (a) follows from the conditional Jensen's inequality, and (b) is due to

$$\mathbb{P}_{N \sim \text{Poi}(\lambda)}(N \text{ is even}) = \sum_{k=0}^{\infty} e^{-\lambda}\frac{\lambda^{2k}}{(2k)!} = e^{-\lambda} \cdot \frac{e^{\lambda} + e^{-\lambda}}{2} \geq \frac{1}{2}.$$

This leads to the claimed regret lower bound in Theorem E.1.

### E.1.2 Lower Bound Analysis for Algorithm 1

Similar to the lower bound analysis for Algorithm 2, we also split into the cases where $n$ is large and $n$ is small, respectively. Recall that for Algorithm 1, the parameter $n$ is the number of random draws from the uniform distribution for each future time. Our current version of Algorithm 1 sets the parameter to be $n = K = \Theta(\log T/\sigma)$.

**Case I: Large $n$.** We first focus on the case where $n \geq 1/\sqrt{\sigma}$. Consider the same construction of $\mathcal{X}, \mathcal{H}$ and the adversary in Case I of Section E.1.1.

The regret analysis is essentially the same as Section E.1.1. For every $t \leq T/2$, the learner in Algorithm 1 essentially generates $(T - t)n \geq T/(4\sqrt{\sigma})$ uniformly random samples (with replacement) in $\mathcal{X}$. A similar analysis to Lemma E.2 shows that for each $j \in [d]$, with $\Omega(1)$ probability there are $\Omega(\sqrt{T/(d\sigma^{1/2})})$ more $+1$ labels than $-1$ labels within $\mathcal{X}_j$ in the hallucinated samples. Consequently, for $t \leq \min\{T/2, \Omega((dT)^{1/2}\sigma^{-1/4})\}$, two calls of the ERM oracle in Algorithm 1 will return the same hypothesis, and the learner's prediction is always $+1$. Similar to Section E.1.1, these time steps lead to an $\Omega(\min\{T, (dT)^{1/2}\sigma^{-1/4}\})$ regret.

**Case II: Small $n$.** Next we turn to the case where $n < 1/\sqrt{\sigma}$.

Consider the same construction as Case II in Appendix E.1.1. For the performance of Algorithm 1, let $r_j$ be the difference between the number of $+1$ and $-1$ labels in the hallucinated samples with input in $\mathcal{X}_j$. One can check that if $r_j \neq 0$, the learner makes half of the mistakes along the alternating sequence; if $r_j = 0$, the fraction of mistakes becomes $3/4$ (Algorithm 1 cyclically predicts a wrong label and makes a random guess). Consequently, the expected regret of Algorithm 1 is lower bounded by $\Omega(T \cdot \mathbb{P}(r_j = 0))$. To compute the probability $\mathbb{P}(r_j = 0)$, note that $r_j = 2M - N$, with $N \sim \text{Bin}(n(T - t), \sigma/d)$ being the number of observations in $\mathcal{X}_j$ in the hallucinated data, and $M \mid N \sim \text{Bin}(N, 1/2)$. Using a similar argument to Section E.1.1, this probability is lower bounded by $\Omega(\min\{1, \sqrt{d/(n\sigma)}\})$, as desired.

### E.2 Proof of Theorem 5.1

**Theorem 5.1** (Restated). *For $1/\sigma \geq d$, any proper algorithm which only has access to the ERM oracle and achieves a regret $o(\min\{T, \sqrt{T(d/\sigma)^{1/2}}\})$ for any $\sigma$-smoothed online learning problem must have an $\omega(\sqrt{d/\sigma})$ total running time.*

We make the following remarks.

1. First, although the lower bound of the regret and running time in Theorem 5.1 does not match the counterparts of Algorithm 2 in Theorem 3.2, the upper and lower bounds share the same $\Theta(\sigma^{-1/4})$ dependence on $\sigma$. This suggests that the improvement from $\Theta(\sigma^{-1/2})$ to $\Theta(\sigma^{-1/4})$ thanks to Poissonization is not superfluous and might be fundamental. We also conjecture that for all efficient algorithms with runtime $\mathrm{poly}(T, d, 1/\sigma)$, the $\Theta(\sigma^{-1/4})$ dependence is the best one can hope for in the regret of such algorithms, as opposed to the $\Theta(\sqrt{\log(1/\sigma)})$ dependence in the statistical regret.

2. Second, Theorem 5.1 shows a $\mathrm{poly}(d, 1/\sigma)$ computational lower bound to achieve the statistical regret $\widetilde{O}(\sqrt{Td\log(1/\sigma)})$, while the $\varepsilon$-net argument in [HRS22] requires a $\mathrm{poly}(\sigma^{-d})$ computational time. One may wonder whether this exponential dependence on $d$ is in fact unavoidable, and this is a missing feature not covered in [HK16]. This motivates the open question in Section 5:

   **Open Question.** *For $d/\sigma \gg T^2$ in the smoothed setting, does any algorithm achieving $o(T)$ regret require $\Omega(\mathrm{poly}(T, 2^d, 1/\sigma))$ computational time given access to the ERM oracle?*

The proof of Theorem 5.1 uses a similar idea to [HK16]. There are two lower bound arguments in [HK16]: one reduces the problem to the Aldous' problem, and the other is based on an explicit construction of the hard instance. Although both arguments could work for our problem, we adopt the latter which corresponds to Theorem 25 of [HK16]. In the sequel, we will always take the domain size $|\mathcal{X}| = 1/\sigma$ so that the smooth adversary becomes the usual adaptive adversary. In the next subsections, we first prove the theorem for the simpler case $d = 1$, and then generalize our argument for any VC dimension $d$.

### E.2.1  The case $d = 1$.

We first show how the argument in [HK16] proves the claimed $\omega(\sqrt{|\mathcal{X}|})$ computational lower bound when $T = \sqrt{|\mathcal{X}|} = \sqrt{1/\sigma}$. Assuming that $N \triangleq \sqrt{|\mathcal{X}|}$ is an integer, we partition the domain $\mathcal{X}$ into disjoint subsets $\mathcal{X}_1, \cdots, \mathcal{X}_N$, each of size $N$. For each $x \in \mathcal{X}$, we associate two independent Rademacher variables $\varepsilon(x)$ and $\varepsilon^\star(x)$, and they are mutually independent across different $x \in \mathcal{X}$. For each $i \in [N]$, the adversary chooses $x_i^\star \sim \mathcal{U}(\mathcal{X}_i)$, and sets the hypothesis class $\mathcal{H} = \{h_x\}_{x \in \mathcal{X}}$ with

$$h_x(x') = \begin{cases} \varepsilon^\star(x') & \text{if } x = x_i^\star, x' = x_j^\star, \text{ and } i \geq j, \\ \varepsilon(x') & \text{otherwise.} \end{cases}$$

At each time $t \in [N]$, the adversary sets $x_t = x_t^\star$, and $y_t = h_{x_N^\star}(x_t) = \varepsilon^\star(x_t^\star)$. Under this setting, [HK16] proved the following lower bound.

**Theorem E.3** (Theorem 25 of [HK16], restated). *Given access to the ERM oracle, any proper algorithm achieving an expected regret at most $N/4$ requires $\Omega(N) = \Omega(\sqrt{|\mathcal{X}|})$ running time.*

Here by running time, we assume that each oracle call takes unit time, and maintaining each element in the input $\{(x_i, y_i)\}_{i \in I}$ to the oracle also takes unit time. We also sketch the proof idea of Theorem E.3 for completeness: the crucial observation is that, when the learner feeds the input $\{(x_i, y_i)\}_{i \in I}$ to the ERM oracle, the oracle can always return some $h \in \{h_0, h_{x_1^\star}, \cdots, h_{x_j^\star}\}$, where $h_0$ is any hypothesis in $\mathcal{H} \setminus \{h_{x_1^\star}, \cdots, h_{x_N^\star}\}$, and $j \in [N]$ is the largest index such that $x_j^\star \in \{x_i\}_{i \in I}$. See Lemma 27 of [HK16] for a proof. Therefore, the label $y_t = \varepsilon^\star(x_t^\star)$ at time $t$ will look random to the learner *unless* the learner has seen a function $h_{x_s^\star}$ for some $s \geq t$. By the above observation, this occurs only if the learner has set one (or more) of $\{x_s^\star\}_{s \geq t}$ as the input to the ERM oracle, but this requires one to find a random element in a size-$N$ set and thus take $\Omega(N)$ time (note that a proper algorithm only observes $\{x_1^\star, \cdots, x_{t-1}^\star\}$ at time $t$). Consequently, with $o(N)$ running time, the learner suffers from an $\Omega(N)$ loss with high probability, while the best expert incurs zero loss - giving the $\Omega(N)$ regret.

Since the restriction of $\mathcal{H}$ on any two elements $\{x, x'\}$ with $x < x'$ could only be one of the three possibilities: $\{(\varepsilon(x), \varepsilon(x')), (\varepsilon^\star(x), \varepsilon(x')), (\varepsilon^\star(x), \varepsilon^\star(x'))\}$, the VC dimension of $\mathcal{H}$ is 1. Therefore, Theorem E.3 gives a valid proof of Theorem 5.1 when $d = 1$ and $T = \sqrt{1/\sigma}$. For $T < \sqrt{1/\sigma}$, the

above construction still gives the $\Omega(T)$ regret lower bound given $o(\sqrt{|\mathcal{X}|})$ computational time. For general $T > \sqrt{1/\sigma}$, we make the following modification to the adversary: partition the time horizon $[T]$ into $N$ intervals $T_1, \cdots, T_N$, each of length $T/N$. For each $i \in [N]$ and $t \in T_i$, the adversary sets $x_t = x_i^\star$, and

$$y_t = \begin{cases} h_{x_N^\star}(x_t) & \text{with probability } \frac{1}{2} + \delta, \\ -h_{x_N^\star}(x_t) & \text{with probability } \frac{1}{2} - \delta. \end{cases}$$

Consequently, the best expert $h_{x_N^\star}$ incurs an expected cumulative loss $(1/2-\delta)T$. Meanwhile, as long as the learner cannot distinguish the distributions $\text{Bern}(1/2 + \delta)^{\otimes(T/N)}$ and $\text{Bern}(1/2 - \delta)^{\otimes(T/N)}$, she is not able to estimate $\varepsilon^\star(x_i^\star)$ based on labels $\{y_t\}_{t \in T_i}$ in the $i$-th interval. This condition is fulfilled when $\delta \asymp \sqrt{N/T}$. In addition, a similar argument for Theorem E.3 shows that with an $o(N)$ computational time, the learner cannot predict future $x_s^\star$ either. Therefore, any proper learner with $o(N) = o(\sqrt{|\mathcal{X}|})$ computational time must incur a regret $\Omega(\delta T) = \Omega(\sqrt{T|\mathcal{X}|^{1/2}})$, which is precisely the statement of Theorem 5.1 for $d = 1$.

### E.2.2   General $d$.

In this section we lift the hypothesis construction for $d = 1$ to general $d$. Since $1/\sigma \geq d$, we assume that $1/(\sigma d)$ is an integer. Partition $\mathcal{X} = \cup_{j=1}^d \mathcal{X}_j$ each of size $|\mathcal{X}|/d$, we apply the hypothesis class $\mathcal{H}$ in the previous section to each $\mathcal{X}_j$, and set the entire hypothesis class as

$$\mathcal{H}_d = \left\{ h = (h_1, \cdots, h_d) \in \mathcal{H}^d : h|_{\mathcal{X}_j} = h_j, \forall j \in [d] \right\}.$$

Clearly the VC dimension of $\mathcal{H}_d$ is $d$. The adversary is constructed as follows: partition $[T]$ into $d$ sub-intervals $T_1, \cdots, T_d$, each of size $T/d$. For the $i$-th sub-interval, we run the subroutine in the previous section independently on $\mathcal{X}_i$. Now suppose that the total runtime is $o(\sqrt{d|\mathcal{X}|})$, then for at least half of the sub-intervals, the runtime during each such interval is $o(\sqrt{|\mathcal{X}|/d})$. By the lower bound for $d = 1$, the expected regret during each such sub-interval is

$$\Omega\left( \min\left\{ \frac{T}{d}, \sqrt{\frac{T}{d} \cdot \left(\frac{|\mathcal{X}|}{d}\right)^{1/2}} \right\} \right) = \Omega\left( \min\left\{ \frac{T}{d}, \sqrt{T \cdot \left(\frac{|\mathcal{X}|}{d^3}\right)^{1/2}} \right\} \right).$$

Summing over at least $d/2$ such independent sub-problems, the total regret lower bound is then $\Omega(\min\{T, \sqrt{T(d|\mathcal{X}|)^{1/2}}\})$, establishing the claim of Theorem 5.1.

## F   Statistical Upper Bound for Real-valued Labels

In this section, we present a statistical upper bound achieved by a computationally inefficient algorithm. The $\mathscr{Q}$ be the algorithm that runs Hedge on a finite subset $\mathcal{H}'$ on $\mathcal{H}$, where $\mathcal{H}'$ is a $\epsilon$-cover of $\mathcal{H}$ with respect to the uniform distribution $\mathcal{U}(\mathcal{X})$. The regret upper bound of this algorithm is bounded as follows.

**Theorem F.1** (Statistical Upper Bound). *For any $\sigma$-smooth adversary $\mathscr{D}_\sigma$, the algorithm $\mathscr{Q}$ described above has regret upper bound*

$$\mathbb{E}[\text{REGRET}(T, \mathscr{D}_\sigma, \mathscr{Q})] \in \widetilde{O}\left( \sqrt{Td\log\left(\frac{T}{d\sigma}\right)} + Gd\log\left(\frac{T}{d\sigma}\right) \right).$$

*Proof.* Let $\mathcal{H}'$ be the smallest $\epsilon$-cover of $\mathcal{H}$ with respect to the uniform distribution, i.e., for any $h \in \mathcal{H}$, there exists a proxy $h' \in \mathcal{H}'$ such that $\mathbb{E}_{x \sim \mathcal{U}(\mathcal{X})}\left[|h(x) - h'(x)|\right] \leq \epsilon$. By lemma B.2, the size of $\mathcal{H}$ can be upper bounded in terms of the pseudo dimension $d$:

$$\log(|\mathcal{H}'|) = \log \mathcal{N}(\epsilon, \mathcal{H}, L_1(\mathcal{U}(\mathcal{X}))) \leq O\left( d\log\left(\frac{1}{\epsilon}\right) \right).$$

Based on the net $\mathcal{H}'$, we also define function class $\mathcal{G}$ as follows.

$$\mathcal{G} = \left\{ g_{h,h'}(x) = |h(x) - h'(x)| : h \in \mathcal{H}, h' \in \mathcal{H}' \text{ is its proxy.} \right\}$$

Letting $L(h, s_{1:T}) = \sum_{t=1}^{T} l(h(x_t), y_t)$ for all $h \in \mathcal{H}$, we now consider the following regret decomposition:

$$\mathbb{E}[\text{REGRET}(T)] = \mathbb{E}\left[ \sum_{t=1}^{T} l(\widehat{y}_t, y_t) - \inf_{h \in \mathcal{H}} L(h, s_{1:T}) \right]$$

$$= \mathbb{E}\left[ \sum_{t=1}^{T} l(\widehat{y}_t, y_t) - \inf_{h' \in \mathcal{H}'} L(h', s_{1:T}) \right] + \mathbb{E}\left[ \inf_{h' \in \mathcal{H}'} L(h', s_{1:T}) - \inf_{h \in \mathcal{H}} L(h, s_{1:T}) \right]$$

Note that the first term is precisely the regret of Hedge on the cover $\mathcal{H}'$. It is thus bounded by

$$\mathbb{E}\left[ \sum_{t=1}^{T} l(\widehat{y}_t, y_t) - \inf_{h' \in \mathcal{H}'} L(h', s_{1:T}) \right] \leq O\left( \sqrt{T \log |\mathcal{H}'|} \right) \in O\left( \sqrt{T d \log\left(\frac{1}{\epsilon}\right)} \right).$$

As for the second term, we reformulate it in terms of class $\mathcal{G}$:

$$\mathbb{E}\left[ \inf_{h' \in \mathcal{H}'} L(h', s_{1:T}) - \inf_{h \in \mathcal{H}} L(h, s_{1:T}) \right] = \mathbb{E}\left[ \sup_{h \in \mathcal{H}} \inf_{h' \in \mathcal{H}'} \sum_{t=1}^{T} l(h'(x_t), y_t) - l(h(x_t), y_t) \right]$$

$$\overset{(a)}{\leq} \mathbb{E}\left[ \sup_{h \in \mathcal{H}} \inf_{h' \in \mathcal{H}'} \sum_{t=1}^{T} G|h(x_t) - h(x_t)| \right] = G \cdot \mathbb{E}_{\mathscr{D}}\left[ \sup_{g \in \mathcal{G}} \sum_{t=1}^{T} g(x_t) \right], \tag{13}$$

where (a) is because the loss function $l$ has Lipschitz constant $G$. Analogous to [HRS22, Claim 3.4], we apply the coupling argument in Lemma B.1 to replace the adaptive sequence $x_t$s by $z_{t,k}$s that are sampled independently from the uniform distribution. Thus we obtain

$$\mathbb{E}_{\mathscr{D}}\left[ \sup_{g \in \mathcal{G}} \sum_{t=1}^{T} g(x_t) \right] \leq T^2(1-\sigma)^K + \mathbb{E}_{\mathcal{U}(\mathcal{X})}\left[ \sup_{g \in \mathcal{G}} \sum_{t=1}^{T} \sum_{i=1}^{K} g(z_{t,k}) \right].$$

The expected supremum can be further bounded in terms of the magnitude of $\mathcal{G}$ (i.e., $\epsilon$) as well as the pseudo dimension of the original hypothesis class $\mathcal{H}$. Using the bound in Lemma F.2, and together with Equation (13), we obtain

$$\mathbb{E}[\text{REGRET}(T)] \leq \widetilde{O}\left( \sqrt{T d \log\left(\frac{1}{\epsilon}\right)} + G\left( T^2(1-\sigma)^K + TK\epsilon + \sqrt{TK\epsilon d \log\left(\frac{1}{\epsilon}\right)} \right) \right).$$

In order to satisfy the condition on $n$ in lemma F.2 and to make the failure probability of the coupling argument sufficiently small, we take $\alpha = 10 \log(T)$, $K = \frac{\alpha}{\sigma}$, $\epsilon = \Theta\left( \frac{d\sigma}{T \log(T)} \log\left( \frac{T \log(T)}{d\sigma} \right) \right)$. With this choice of parameters, we have $T^2(1-\sigma)^K = o(1)$ and

$$\mathbb{E}[\text{REGRET}(T)] \leq O\left( \sqrt{T d \log\left(\frac{1}{\epsilon}\right)} + G\left( \frac{T \log(T)}{\sigma} \epsilon + \sqrt{\frac{T \log(T)}{\sigma} \epsilon d \log\left(\frac{1}{\epsilon}\right)} \right) \right)$$

$$\leq \widetilde{O}\left( \sqrt{T d \log\left(\frac{T}{d\sigma}\right)} + G d \log\left(\frac{T}{d\sigma}\right) \right),$$

as desired. $\qquad \square$

**Lemma F.2** (Concentration for the expected value of supremum). *When $n \geq \Omega\left(\frac{d}{\epsilon}\log\left(\frac{1}{\epsilon}\right)\right)$, we have*

$$\mathop{\mathbb{E}}_{x_{1:n} \overset{iid}{\sim} \mathcal{U}(\mathcal{X})} \left[ \sup_{g \in \mathcal{G}} \sum_{i=1}^{n} g(x_i) \right] \leq O\left( n\epsilon + \sqrt{n\epsilon d \log\left(\frac{1}{\epsilon}\right)} \right).$$

*Proof.* We will use the bound on expected values of suprema of empirical processes in [GK06, Theorem 3.1]. To apply their result, the first step is to establish a bound on the $L_2(P)$-covering number of class $\mathcal{G}$. Let $P_n = \frac{1}{n}\sum_{i=1}^{n} \delta_{x_i}$ be the empirical distribution based on independent samples $x_1, \cdots, x_n$. A similar argument to [BKP97, Lemma 2] gives us

$$\mathcal{N}(\epsilon, \mathcal{G}, L_2(P_n)) \leq \mathcal{N}(\frac{\epsilon}{2}, \mathcal{H}, L_2(P_n))^2.$$

Thus we obtain

$$\log \mathcal{N}(\epsilon, \mathcal{G}, L_2(P_n)) \leq 2\log \mathcal{N}(\frac{\epsilon}{2}, \mathcal{H}, L_2(P_n)) \leq 2\log \mathcal{M}(\frac{\epsilon}{2}, \mathcal{H}, L_2(P_n)) \leq O\left(d\log(\frac{1}{\epsilon})\right),$$

where $\mathcal{M}$ denotes the packing number and the last inequality is due to [Bar06, Theorem 3.1]. Therefore, for the function $H(x) = O(d\log x)$, we can guarantee that for any $\epsilon > 1$,

$$\log \mathcal{N}(\epsilon, \mathcal{G}, L_2(P_n)) \leq H(1/\epsilon),$$

satisfying the condition of [GK06, Theorem 3.1]. Therefore, when $n \geq \Omega\left(\frac{H(1/\epsilon)}{\epsilon}\right) = \Omega\left(\frac{d}{\epsilon}\log\left(\frac{1}{\epsilon}\right)\right)$, [GK06] gives us

$$\mathop{\mathbb{E}}_{\mathcal{U}} \left[ \sup_{g \in \mathcal{G}} \sum_{i=1}^{n} \left(g(x_t) - \mathbb{E}[g(x_t)]\right) \right] \leq O\left(\sqrt{n\epsilon H(1/\epsilon)}\right) = O\left(\sqrt{n\epsilon d \log\left(\frac{1}{\epsilon}\right)}\right).$$

Finally, since $\mathbb{E}_{\mathcal{U}} g(x) \leq \epsilon$ for any $g \in \mathcal{G}$, we obtain

$$\mathbb{E} \left[ \sup_{g \in \mathcal{G}} \sum_{i=1}^{n} g(x_t) \right] \leq O\left( n\epsilon + \sqrt{n\epsilon d \log\left(\frac{1}{\epsilon}\right)} \right),$$

and the proof is complete. $\qquad\square$

## G  Proof of the Admissible Relaxation Framework

**Proposition 4.1** (Restated). *In the smoothed online learning setting, let $\mathscr{Q} = (\mathcal{Q}_1, \cdots, \mathcal{Q}_T)$ be an algorithm that is admissible with respect to relaxations $\mathbf{Rel}_T(\mathcal{H})$, then the following bound on the expected regret holds regardless of the strategies $\mathscr{D}_\sigma$ of the adversary:*

$$\mathbb{E}[\textsc{Regret}(T, \mathscr{Q}, \mathscr{D}_\sigma)] \leq \mathbf{Rel}_T(\mathcal{H} \mid \emptyset) + O(\sqrt{T}).$$

*Proof of proposition 4.1.* To prove this lemma we break the expected regret into two parts:

$$\mathbb{E}[\textsc{Regret}(T)] = \mathop{\mathbb{E}}_{\mathscr{D},\mathscr{Q}} \left[ \sum_{t=1}^{T} \mathop{\mathbb{E}}_{\widehat{y}_t \sim \mathcal{Q}_t} [l(\widehat{y}_t, y_t)] - \inf_{h \in \mathcal{H}} L(h, s_{1:T}) \right] + \mathop{\mathbb{E}}_{\mathscr{D},\mathscr{Q}} \left[ \sum_{t=1}^{T} l(\widehat{y}_t, y_t) - \mathop{\mathbb{E}}_{\widehat{y}_t \sim \mathcal{Q}_t} [l(\widehat{y}_t, y_t)] \right].$$

For the first part, we use an inductive argument to show that

$$\mathop{\mathbb{E}}_{\mathscr{D},\mathscr{Q}} \left[ \sum_{t=1}^{T} \mathop{\mathbb{E}}_{\widehat{y}_t \sim \mathcal{Q}_t} [l(\widehat{y}_t, y_t)] - \inf_{h \in \mathcal{H}} \sum_{t=1}^{T} l(h(x_t), y_t) \right] \leq \mathbf{Rel}_T(\mathcal{H} \mid \emptyset). \tag{14}$$

According to the definition of admissibility, we have

$$
\mathop{\mathbb{E}}_{\mathscr{D},\mathscr{Q}} \left[ \sum_{t=1}^{T} \mathop{\mathbb{E}}_{\widehat{y}_t \sim \mathcal{Q}_t} [l(\widehat{y}_t, y_t)] \qquad \underbrace{- \inf_{h \in \mathcal{H}} \sum_{t=1}^{T} l(h(x_t), y_t)}_{\leq \mathbf{Rel}_T(\mathcal{H}|s_{1:T}) \text{ by 2nd condition of admissibility}} \right]
$$

$$
\leq \mathop{\mathbb{E}}_{\mathscr{D},\mathscr{Q}} \left[ \mathbb{E} \left[ \sum_{t=1}^{T-1} \mathop{\mathbb{E}}_{\widehat{y}_t \sim \mathcal{Q}_t} [l(\widehat{y}_t, y_t)] + \underbrace{\mathop{\mathbb{E}}_{x_T \sim \mathcal{D}_T} \left[ \mathop{\mathbb{E}}_{\widehat{y}_T \sim \mathcal{Q}_T} [l(\widehat{y}_T, y_T)] + \mathbf{Rel}_T(\mathcal{H} \mid s_{1:T}) \right]}_{\leq \mathbf{Rel}_T(\mathcal{H}|s_{1:T-1}) \text{ by 1st condition of admissibility}} \Bigg| s_{1:T-1} \right] \right]
$$

$$
\leq \mathop{\mathbb{E}}_{\mathscr{D},\mathscr{Q}} \left[ \mathbb{E} \left[ \sum_{t=1}^{T-1} \mathop{\mathbb{E}}_{\widehat{y}_t \sim \mathcal{Q}_t} [l(\widehat{y}_t, y_t)] + \mathbf{Rel}_T(\mathcal{H} \mid s_{1:T-1}) \Bigg| s_{1:T-1} \right] \right]
$$

$$
= \mathop{\mathbb{E}}_{\mathscr{D},\mathscr{Q}} \left[ \sum_{t=1}^{T-1} \mathop{\mathbb{E}}_{\widehat{y}_t \sim \mathcal{Q}_t} [l(\widehat{y}_t, y_t)] + \mathbf{Rel}_T(\mathcal{H} \mid s_{1:T-1}) \right],
$$

where the last step uses the tower property of conditional expectations.

Repeat this process for $(T-1)$ times and note that $\mathbf{Rel}_T(\mathcal{H} \mid \emptyset)$ is a constant that does not dependent on $\mathscr{D}$ proves Equation (14).

Since the second part is the expected sum of a martingale difference sequence, we apply the Azuma-Hoeffding inequality and obtain

$$
\mathop{\mathbb{E}}_{\mathscr{D},\mathscr{Q}} \left[ \sum_{t=1}^{T} l(\widehat{y}_t, y_t) - \mathop{\mathbb{E}}_{\widehat{y}_t \sim \mathcal{Q}_t} [l(\widehat{y}_t, y_t)] \right] \leq \int_0^\infty \exp\left( -\frac{2t^2}{T} \right) dt \in O(\sqrt{T}). \tag{15}
$$

Combining Equation (14) and Equation (15) completes the proof. □

# H  Transductive Learning with $K$ Hints

## H.1  Model

In the traditional transductive setting, the adversary releases the sequence of unlabeled instances $\{x_t\}_{t=1}^{T}$ to the learner before the game starts. We generalize this setting and introduce a $K$-hint version of transductive learning. In this setting, the exact sequence of instances is replaced with a sequence of $K$ *hints* per time step such that the set of hints at each time step includes the instance at that time step. More formally, before the interaction starts, the adversary releases $T$ sets (multisets) of size $K$ to the learner. We denote these sets by $\{Z_t = \{z_{t,1}, \cdots, z_{t,K}\}\}_{t=1}^{T}$. On releasing these sets, the adversary promises to always pick $\mathcal{D}_t$ supported only on the elements of $Z_t$. The regret of a learner with prediction rules $\mathscr{Q}$ on the adaptive sequence $\mathscr{D} = (\mathcal{D}_1, \cdots, \mathcal{D}_T)$ following above restrictions is defined as:

$$
\mathbb{E}[\textsc{Regret}(T, \mathscr{D}, \mathscr{Q})] = \mathop{\mathbb{E}}_{\mathscr{D},\mathscr{Q}} \left[ \sum_{t=1}^{T} l(\widehat{y}_t, y_t) - \inf_{h \in \mathcal{H}} \sum_{t=1}^{T} l(h(x_t), y_t) \right].
$$

## H.2  Efficient Algorithm for Transductive Online Learning with $K$ Hints

We will show an oracle-efficient regret upper bound of $O(\sqrt{dTK})$ by constructing an oracle-efficient algorithm based on the random playout technique. We consider the optimization oracle defined in Definition 2.3 with the loss functions specified by $l^{\mathrm{r}}(\widehat{y}, y) = \frac{1}{2G} l(\widehat{y}, y)$ and $l^{\mathrm{b}}(\widehat{y}, y) = \mathbf{1}\{\widehat{y} \neq y\} - \frac{1}{2}$.

Similar to Algorithm 1, at each time step $t$, our algorithm applies the offline optimization oracle to two input sequences: One where the real history $s_{1:t-1}$ is mixed with two copies[5] of randomly labeled set of all hints corresponding to future time steps and the current instance is labled $+1$, and another, where the current instance is labeled $-1$.

More specifically, with $\mathcal{E}^{(t)} = \{\epsilon_{i,k}^{(t)}\}_{i=t+1:T,k=1:K}$ denoting the set of random labels and $S^{(t)} = (Z_{t+1:T}, \mathcal{E}^{(t)})$ denoting the set of hints labeled by $\mathcal{E}^{(t)}$, we consider

$$\widehat{y}_t = \mathsf{OPT}\left(s_{1:t-1}; S^{(t)} \cup S^{(t)} \cup \{(x_t, -1)\}\right) - \mathsf{OPT}\left(s_{1:t-1}; S^{(t)} \cup S^{(t)} \cup \{(x_t, +1)\}\right). \quad (16)$$

See Algorithm 3 for a formal description of the algorithm.

---

**Algorithm 3:** Oracle-Efficient Online Transductive Learning with $K$ Hints

**Input:** $T, K, \{Z_t\}_{t=1}^T$

1 **for** $t \leftarrow 1$ **to** $T$ **do**
2      Receive $x_t$. Assert that $x_t \in Z_t$
3      **for** $i = t+1, \cdots, T$; $k = 1, \cdots, K$ **do**
4          Draw new $\epsilon_{i,k}^{(t)} \sim \mathcal{U}(\{-1, +1\})$.
5      **end**
6      $S_1^{(t)} \leftarrow \left\{(z_{i,k}^{(t)}, \epsilon_{i,k}^{(t)}), (z_{i,k}^{(t)}, \epsilon_{i,k}^{(t)})\right\}_{\substack{i=t+1:T \\ k=1:K}}$            // Two copies of each tuple
7      $S_2^{(t)} \leftarrow \{(x_\tau, y_\tau)\}_{\tau=1}^{t-1}$
8      $p_t \leftarrow \frac{1}{2} + \frac{\mathsf{OPT}_{\mathcal{H},l}(S_t^{(1)} \cup S_t^{(2)} \cup \{(x_t, +1)\}) - \mathsf{OPT}_{\mathcal{H},l}(S_t^{(1)} \cup S_t^{(2)} \cup \{(x_t, -1)\})}{2}$.
9      With probability $p_t$, predict $\widehat{y}_t = -1$; otherwise predict $\widehat{y}_t = +1$
10      Receive $y_t$, suffer loss $l(\widehat{y}_t, y_t)$.
11 **end**

---

**Theorem H.1** (Regret Bound for Efficient $K$-Hint Transductive Learning). *In the setting of transductive learning with $K$-hints, Algorithm 3 achieve expected regret bound of $O(\sqrt{dTK})$. The algorithm can be implemented using two calls to the optimization oracle per round.*

The proof of Theorem H.1 follows a similar approach to that in [RSS12] and uses the admissible relaxation framework. This proof is also very similar to the proof of Theorem 3.1 for the case of smoothed adversaries. We include the proof of this Theorem for completeness.

Specifically, we will show that the algorithm is admissible with respect to the following relaxation:

$$\mathbf{Rel}_T(\mathcal{H} \mid s_{1:t}) = \mathop{\mathbb{E}}_{\mathcal{E}^{(t)}}\left[\sup_{h \in \mathcal{H}} \left\{ 2G \sum_{\substack{i=t+1:T \\ k=1:K}} \epsilon_{i,k}^{(t)} h(z_{i,k}) - \sum_{i=1}^t l(h(x_i), y_i) \right\} \right], \qquad t = 0, \cdots, T.$$

Using the language of regularized Rademacher complexity introduced in Section 4.3, the relaxation at the end of time step $t$ can be written as the Rademacher complexity for the union of future hints, regularized by the past total loss. That is,

$$\mathbf{Rel}_T(\mathcal{H} \mid s_{1:t}) = 2G \cdot \mathfrak{R}(-L^{\mathrm{r}}(\cdot, s_{1:t}), Z_{t+1:T}), \qquad (17)$$

where $L^{\mathrm{r}}(h, s_{1:t}) = \sum_{i=1}^t l^{\mathrm{r}}(h(x_i), y_i) = \frac{1}{2G} \sum_{i=1}^t l(h(x_i), y_i)$ for $h \in \mathcal{H}$.

To use the relaxation framework and Proposition 4.1, it suffices to establish two claims: 1) the relaxation in Equation (17) is admissible in the $K$-hint setting, 2) the value of this relaxation at the beginning of the game is not too large.

For the second claim, we notice that $\mathbf{Rel}_T(\mathcal{H} \mid \emptyset)$ is equal to the unregularized Rademacher complexity for the dataset that includes all the hints. Since there are at most $TK$ hints, the Rademacher

---

[5]We use two copies to scale the loss appropriately.

complexity is at most $\widetilde{O}(\sqrt{dTK})$ according to Lemma B.3. That's where we get the extra $\sqrt{K}$ in the bounds compared to the standard transductive setting.

The first claim is the more technically interesting one. For admissibility, here we will focus on proving the following bound

$$\sup_{x_t \in Z_t} \sup_{y_t \in \mathcal{Y}} \underbrace{\left\{ \mathbb{E}_{\widehat{y}_t \sim \mathcal{Q}_t} [l^r(\widehat{y}_t, y_t)] + \mathfrak{R}(-L_t^r, Z_{t+1:T}) \right\}}_{(a)} \leq \sup_{x_t \in Z_t} \mathfrak{R}(-L_{t-1}^r, Z_{t+1:T} \cup \{x_t\}) \leq \mathfrak{R}(-L_{t-1}^r, Z_{t:T}),$$

$$(18)$$

where $L_t^r(h)$ abbreviates for $L^r(h, s_{1:t})$.

Let us consider the L.H.S of the above inequality and note that for any fixed $x_t$, the term (a) captures the standard transductive learning setting with $Z_{t+1:T}$ being the set of unlabeled instances for the future. In this case, the convexity of loss function $l^r$ together with the min-max theorem can be used to show that the learner's strategy $\mathcal{Q}_t$, which makes the two values inside the supremum over $\mathcal{Y}$ equalize as $y_t$ takes value $-1$ and $+1$, is indeed the optimal strategy. At a high level, this technique which is also used by [RSS12], gives rise to the algorithm in Equation (16) and proves the first inequality in inequality (18). We refer the readers to [RSS12, Lemma 12] for more details about the proof.

The second transition in Inequality (18) can be established using the fact that regularized Rademacher complexity is monotone in the dataset, as shown in Lemma 4.1. Therefore, we have proved that the relaxation given by Equation (17) is admissible, and the final regret upper bound follows from Proposition 4.1.