# OpenReview forum: "Oracle-Efficient Online Learning for Smoothed Adversaries"
_NeurIPS.cc/2022/Conference — NeurIPS 2022 Accept_

### Official Review · Reviewer_gcMM · 2022-06-17

**Rating:** 6
**Confidence:** 3
**Soundness:** 3 good
**Presentation:** 4 excellent
**Contribution:** 3 good

**Summary:**

This paper considers the oracle-efficiency of smoothed online learning. In this setting, the output distribution of the nature is constrained to have a constant upper bound. Assuming an offline optimization oracle, they give an $O(\sqrt{T d \sigma^{-1}})$ regret bound for real valued loss and an $O(\sqrt{T d \sigma^{-1/2}})$ regret bound for binary valued loss as their main results. The algorithms achieving these regret bounds are based on FPL.

**Questions:**

Can you give more justification on the smoothed analysis setting? Or any concrete example?

**Limitations:**

Yes

**Strengths And Weaknesses:**

The results are sound and solid, and the algorithms and proof ideas are very intuitive. The Poissonization technique is novel and may be of independent interest.

The major concern I have is on the setting of smoothed online learning. There are many ways to characterize online learning beyond worst-case, and smoothed online learning doesn't seem natural to me. I don't see any practical scenario supporting the smoothed setting, and in theory there are other more intuitive ways: for example assuming noise or predictable sequences. The way that smoothed analysis restricts the distribution of the nature's outputs doesn't make much sense and the constant upper bound seems a very strong assumption.

The improvement over regret bounds of classical online learning doesn't seem significant: the dominating term $O(\sqrt{T})$ isn't improved. One may expect a more significant improvement from the strong additional assumption. A (not very relevant) example is that one can improve the regret bound to $O(\log T)$ assuming an abstention option.

The paper is well-written and easy-to-follow. It spells out the assumptions, algorithms and intuitions clearly.

Though the results of this paper are solid, due to the concurrent work of BDGR22 which is already accepted at COLT2022, there doesn't seem to be enough improvement and novelty upon it. For this reason (along with my concern on the smoothed analysis setting), unfortunately I tend to reject this paper.

---

> ### Author Response · Authors · 2022-08-02
> **Response to Reviewer gcMM: Comparison and Treatment of Concurrent Works**
>
> We disagree that there is not enough improvement and novelty upon the concurrent work of BDGR22. Moreover, we disagree with the reviewer’s implicit assessment that our work should be treated as a follow up to BDGR22 rather than a concurrent work.
>
> **Comparison with BDGR22**: First, our results are stronger in terms of both the regret dependence on smoothness parameters and the number of oracle calls required in each time step. In the binary classification setting, our Poissionizationed FTPL algorithm achieves regret $O(\sqrt{Td\sigma^{-1/2}})$, while their algorithm only obtains the worse bound of $O(\sqrt{Td\sigma^{-1}})$. For the case of real-valued functions, we obtain regret of order using $O(\sqrt{Td\sigma^{-1}})$ using 2 oracle calls per round, while their corresponding relaxation-based algorithm only achieves regret $O(\sigma^{-1}\sqrt{Td})$ with $O(T)$ calls per round. Subsequent to when the first versions of both papers were made available online, BDGR22 improved their relaxation technique and now demonstrate the same regret bound as ours, while still using more oracle calls. BGDR22 also developed an FTPL-based algorithm for real-valued functions that calls the oracle only once per round, but the regret bound of this algorithm is $O(T^{2/3}\sigma^{-1/3})$, which is significantly suboptimal in $T$.
>
> Second, our paper uses a different and novel approach. In order to control the stability of FTPL in the binary setting, we use the novel Poissonization technique, which gives us an additional degree of independence and allows us to use information-theoretic tools. More importantly, we introduce an entirely new framework, called modified generalization error, which takes a different perspective on generalization and stability analysis of FTPL-type algorithms. It looks into the generalization error of ERM classifiers when they are trained on uniformly generated training samples and tested on smoothly distributed fresh instances. We believe it will be of independent interest for transductive and other settings of beyond the worst-case learning.
>
> **On treating concurrent works in review processes**: Setting aside the fact that our results are stronger, novel, and of independent interest (even conditioned on the publication of BDGR22), the publication of the concurrent works in other conferences is only evidence that the community is very interested in this line of work. Our work has been conducted concurrently and independently from BDGR22 which is evident by the fact that the first versions of these papers were made public within days of each other. We urge the reviewer to not treat concurrent works as follow-ups to one another. We believe doing so will greatly harm the review process and the community at large by encouraging treating publication as competition. Our work and BDGR22 have independent insights and contributions and their publication will draw independent readership and interest from the field.

---

> ### Author Response · Authors · 2022-08-02
> **Response to Reviewer gcMM: With Regard to Improved Regret Bounds**
>
> Our improvement in regret is significant: Note that without the smoothness assumption, even any hypothesis class that is expressive enough to contain thresholds in one dimension is unlearnable (due to infinite Littlestone dimension), which means that no algorithm can get regret of o(T). However, our algorithms show that, under mild smoothness assumption, these classes become statistically and computationally learnable. This is a significant improvement because the dominating term decreases from $\Theta(T)$ to $O(\sqrt{T})$.
>
> **Why we should not expect a better than sqrt{T} regret using assumptions similar to that of Smoothed Analysis**: The primary feature of Smoothed Analysis in online learning is to restrict the harmful correlations that an adaptive adversary can cause over time. ven if the adversary is completely removed and replaced by full stochasticity, e.g., in an extreme case where instances come from a fixed distribution that is uniform over the domain, the regret of $\Theta(\sqrt{T})$ is tight in an information-theoretic sense. This is a well-established result in PAC learning theory. Therefore, a further improvement on the $T$ term cannot be hoped for.
>
> **General notions of beyond worst-case analysis**:  In some sense, we can think of smoothness as a stochastic version of predictable sequences. This is because the coupling argument guarantees that with high probability, any sequence of $T$ instances generated by adaptive smoothed adversaries can be seen as a subset of $T/\sigma$ uniformly random instances. This formulation is especially useful if we want to work on general spaces while not making any specific assumption on a sequence that depends heavily on the particular representation. Furthermore, it is not clear if one can define notions such as predictability of sequences in representation-independent ways and essentially all previous work in that space use hypothesis classes on normed spaces.

---

> ### Author Response · Authors · 2022-08-02
> **Response to Reviewer gcMM: On the justification of Smoothed Analysis**
>
> **Standard beyond the worst-case model in algorithm design** There is a long and rich line of work in TCS and machine learning that uses smoothed analysis to justify the empirical performance of algorithms that are used in practice. A celebrated example of this is the analysis of the simplex algorithm by Spielman and Teng. This notion has gone on to be influential in various areas such as combinatorial optimization and game theory. See [Part 4, Roughgarden 2021] for a more detailed look into smoothed analysis. In particular, the specific notion of smoothness we use in online learning has been around at least since [Rakhlin, Sridharan, &Tewari, 2011], and has played a key role in several other important works [Gupta & Roughgarden 2017, Balcan et. al, 2021, Cohen-Addad & Kanade 2017, Haghtalab, Roughgarden & Shetty 2020-2021, etc]. Using an upper bound on the density of the distribution is the modern (and more general) interpretation of smoothed analysis. Please refer to [Part 4, Roughgarden 2021] for more details.
>
> **The value of our work**: What is unique and interesting about our work is not the introduction of smoothed analysis, but rather enabling computationally efficient algorithms to exist. We fulfill the promise of smoothed analysis in online learning by establishing that we simultaneously get improved regret bounds and efficient algorithms under smoothness.
>
> **Why smoothed analysis is particularly compelling for online learning**:
> Smoothness generalizes various notions of noise, because the distribution of the perturbed instances is a convolution with the noise distribution. For example, in R^n, if we consider Gaussian noise, it is easy to check that the input at each time step then becomes smooth (i.e. with bounded density that depends on dimension $n$ and standard deviation).  In fact, the reason that noise makes this setting learnable is because it introduces anti-concentration to the input distribution, which is exactly what smoothness aims to capture.
>
> Smoothness is a general notion that works equally well in both combinatorial and continuous (geometric) settings, without having to worry about details of the noise generating process. Moreover, the smoothness parameter $\sigma^{-1}$ need not be constant and can depend on the problem parameters such as dimension (our algorithms still get non-trivial regret in such settings), and can even depend sublinearly on $T$. In the online binary classification setting, our bound in Theorem 3.2 shows that efficient learning is possible even when the smoothness parameter is linear in $T$, which is exactly the case of transductive online learning. Therefore, we believe that working in this general setting allows us to design algorithms that are robust to assumptions about noise.

---

> ### Author Response · Authors · 2022-08-02
> **Response to Reviewer gcMM**
>
> We thank the reviewer for the comments. However, we disagree with the reviewer on the importance of smoothed analysis and its characterization. We also address the comparisons with concurrent works.

---

> > ### Comment · Reviewer_gcMM · 2022-08-04
> > **Response to the authors**
> >
> > About technical stuff: thank you for the very detailed response! Though the assumption still feels strong to me, I agree it's an established notion and really appreciate the introduction of the history of smoothed analysis, and why it makes sense in online learning. As you mentioned that "In the online binary classification setting, our bound in Theorem 3.2 shows that efficient learning is possible even when the smoothness parameter is linear in $T$, which is exactly the case of transductive online learning.", it would be great if you can recover the regret bound in the classical online learning setting using smooth analysis (I believe one can approach the $\sqrt{T}$ rate by smoothing the domain and use a $\sigma$ with dependence on $T$), which would be a very convincing argument supporting the assumption.
> >
> > About comparison with BDGR22: this was my minor concern, I was very aware it was a concurrent work and I had no "implicit" intent to treat it as a follow-up. The fact is, there's already a published paper with very similar results. Whether it was a concurrent work or not, now the significance of the overlapping part somewhat degrades, which one can't pretend to ignore. I understand this situation (having a concurrent work with worse results accepted) is upsetting and irritating, but I feel it's offensive to imply subjective opinions of a reviewer from a few sentences.

---

> > > ### Author Response · Authors · 2022-08-09
> > > **Response to Review gcMM: about the overlapping part with the concurrent work**
> > >
> > > **In response to “the significance of the overlapping part somewhat degrades”:** We believe that the overlap with concurrent results are minor, on the other hand, there are several results and techniques that are unique to our work and have significant implications. Above, we highlighted two of the important consequences of the fact that our work achieves a regret of $\min\left(\sqrt{dT\sigma^{-1 / 2}},\sqrt{T(d|\mathcal{X}|)^{1/2}}\right)$. Furthermore, these stronger results and consequences are only enabled by the novel techniques we introduced in this work, i.e., the modified generalization error and poissonization techniques. We believe these techniques are of independent interest and will impact the field more broadly.

---

> > > ### Author Response · Authors · 2022-08-09
> > > **Response to Review gcMM: about the connection with classical settings**
> > >
> > > Thank you for the response.
> > >
> > > **In response to “it would be great if you can recover the regret bound in the classical online learning setting using smooth analysis …”:** Our techniques indeed recover and improve upon several classical results in online learning. We will restate these below. We also emphasize that the connection to these classical settings is unique to our work — existing and concurrent works in this space are not capable of achieving these results, in part due to the weaker nature of their results.
> > >
> > > 1. **Classical Online Learning with Finite domain**: In the finite domain case, our Poissonized FTPL algorithm achieves a regret of $\sqrt{T(d|\mathcal{X}|)^{1/2}}$, which improves upon the state-of-the-art regret bound of $\sqrt{T|\mathcal{X}|}$ for oracle-efficient by [DS16], because the VC dimension $d$ is bounded by $|\mathcal{X}|$ in the worst case and can be significantly smaller for most hypothesis classes.
> > > 2. **Classical Transductive Learning**: In the transductive learning setting, our algorithms recover and improve upon the best known results. In particular,
> > >     -  **For the relaxation-based algorithm (Algorithm 1)**, as discussed in Section 4.3 of our paper, the relaxation we use to design this algorithm is a more general form of the relaxation that characterizes the standard transductive learning setting. Therefore, if we directly set $S^{(t)}$ to be the union of unlabeled instances for the future rather than a set of uniformly sampled instances from the base distribution, our Algorithm 1 reduces to the standard transductive learning algorithm proposed by [RSS12], and achieves the regret of $\sqrt{Td}$.
> > >      -  **Poissonized FTPL algorithm (Algorithm 2)** can be combined with the observation that transductive learning is equivalent to having a finite domain of size $|\mathcal{X}|=T$. Using our $\sqrt{T(d|\mathcal{X}|)^{1/2}}$ regret bound for finite domains, we achieve a $T^{3/4}d^{1/4}$ regret bound for transductive setting. This improves upon the regret $T^{3/4}d^{1/2}$ of [KK06] by improving the dependence on the VC dimension. This improvement is significant for several reasons. On the historical front, up to now the FTPL approach for solving transductive learning was considered to be inferior to the RSS12’s relaxation-based approach since they shared the same $d^{1/2}$ dependence but FTPL had a worse dependence on $T$. Our work shows that these two approaches both have their strengths, since Poissonized FTPL demonstrates a better dependence on $d$ compared to RSS12’s relaxation-based approach. This is also welcomed news for applications where the “proper-learning” aspect of FTPL is in particular important.
> > > The improvements above are only possible because our novel techniques and approach are capable of achieving the very strong regret bound of Theorem 3.2 with $\min\left(\sigma^{-1/4}\sqrt{dT}, \sqrt{T(d|\mathcal{X}|)^{1/2}}\right)$ dependence. The regret bounds with dependencies of $\sigma^{-1/2}\sqrt{dT}$ and $\sigma^{-1}\sqrt{dT}$, such as those in [BDGR22], are not capable of improving upon the existing results, or as in the case of [DS16], recovering these bounds.
> > >
> > > 3. The aforementioned results are highlighted in Table 1 and Cor 3.3 for more detail. We will expand on the discussion of Cor 3.3 in the final version of the paper to emphasize the significance of these results more.

---

> > > > ### Comment · Reviewer_gcMM · 2022-08-09
> > > > **Response to the authors**
> > > >
> > > > Thank you for your detailed response on the connection with classical settings ! I think explaining how your results recover/improve regret bounds in classic settings is very important: it convinces audiences that the smoothed analysis setting is legit, and I believe it's worth using a separate sub-section discussing these (the current response is already good with some polishing).
> > > >
> > > > About the overlapping part: this was my minor concern. Now that you have addressed my main concern, I will raise my score accordingly.

---

### Official Review · Reviewer_dbrU · 2022-07-07

**Rating:** 7
**Confidence:** 2
**Soundness:** 4 excellent
**Presentation:** 3 good
**Contribution:** 3 good

**Summary:**


This paper considers online learning with smoothed adversaries, which is a special case of online learning. Online learning is a sequential setting in which in round $t = 1, \ldots, T$ a learner issues a prediction, suffers a loss corresponding to that prediction, and subsequently observes some feedback. The goal is to control the regret, which is the difference between the cumulative losses of the learner and the cumulative losses of the best fixed prediction in hindsight. Normally, the losses are chosen by an adversary or sampled from a distribution. However, in this paper the authors consider a setting in which the adversary generates losses from a distribution that is bounded by $1/\sigma$ times the uniform distribution.

Non-efficient algorithms for this setting obtain regret bounds of order $\sqrt{dT\ln(\sigma^{-1})}$, where $d$ is a problem specific parameter. The authors provide an efficient algorithm with $\sqrt{d T \sigma^{-1}}$ regret for real or binary losses and an algorithm with $\sqrt{dT\sigma^{-1/2}}$ for binary losses.

Lower bounds for any efficient algorithm are of order $\sqrt{d^{1/2}T\sigma^{-1/2}}$ and for algorithms in the same class of the algorithms that are used to obtain the new upper bounds are of order $\sqrt{dT\sigma^{-1/2}}$.


**Questions:**

See the weaknesses.

**Limitations:**

The authors have adequately addressed the limitations and potential negative societal impact of their work

**Strengths And Weaknesses:**

Strengths

The paper is well written.

The techniques to obtain the upper bound are nice and could prove useful beyond this paper.

There are immediate consequences beyond the setting at hand: the regret bounds imply improvements in worst-case binary classification in terms of the domain size and in transductive binary classification.

The paper is well contextualised.

The jump in regret bounds from inefficient to efficient algorithms is explained by the lower bounds, although it is not clear whether these lower bounds are tight.



Weaknesses

The setting seems to be a natural interpolation between adversarially and stochastically generated losses. However, it also seems to be quite a strong assumption to assume knowledge of $\sigma$. Is there a reason why in general one can assume knowledge of $\sigma$?

The algorithm needs knowledge of $\sigma$ to draw samples from $1/\sigma$ times the uniform distribution, which makes me believe that even having a slightly inaccurate $\sigma$ would lead to profound consequences for the regret bound. Naively trying to learn $\sigma$ choosing a grid of possible $\sigma$ and then running an expert algorithm seems quite costly in terms of oracle calls. Is there a more clever approach?

---

> ### Author Response · Authors · 2022-08-02
> **Response to Reviewer dbrU**
>
> We thank the reviewer for the positive review. Below we answer your questions.
>
> **Unknown the value of sigma**: Please see our general comments for a detailed explanation. In particular, we do not need to know the exact value of sigma.

---

### Official Review · Reviewer_4m7m · 2022-07-08

**Rating:** 9
**Confidence:** 3
**Soundness:** 4 excellent
**Presentation:** 3 good
**Contribution:** 4 excellent

**Summary:**

This work provides so called oracle efficient algorithms for online learning, in the smoothed adversarial setting. This problem corresponds to: given access to an ERM oracle, to design an algorithm whose regret in the worst-case regret under distributions which are $\sigma$-smooth, is as small at possible. The results of the paper provide $O(\sqrt{T})$ regret algorithms in this setting, where also the noise parameter $\sigma$, as well as the VC dimension $d$, appear in the bounds. The paper focuses in the setting of convex Lipschitz loses and binary cases separately. For the former, a relaxation framework (a technique tracing back to [RSS12]) is proposed, which also leverages a coupling technique that exploits the smoothness of the adversary to establish that any smooth sample will be contained who on a larger sample of iid uniform random variables. For the binary case, tighter bounds are obtained via a follow the perturbed leader (FTPL) approach, which also leverages the aforementioned coupling. Several other ideas, regarding the use of a form of regularized Rademacher complexity, stability analysis, etc.; are used. The paper also provides some computational lower bounds, showing that for small values of $\sigma$, the improved regret bounds of previous work necessarily require polynomial running time in $1/\sigma$.

**Questions:**

1. Can the authors add some references on transduction learning, just to point interested readers into relevant literature, to better grasp the results?
2. Page 15, equation (5). ${\cal Q}_t$ is used as notation in the equations before it is introduced. I suggest to mention this before the chain of equations, rather than after.
3. Page 17, line 568. What is $I$? Also I am inferring (but it is not clear from the writing) that ${\cal E}=(\epsilon_1,\ldots,\epsilon_I)$?
4. Page 22, Lemma C.6. It was unclear for me (until reading the proof), that $X_I$ is being conditioned on $E,X_{\setminus I}$. I would suggest writing it instead as $X_I | (E,X_{\setminus I})$.

**Limitations:**

Yes, I think limitations of the paper are discussed, and it even points out to an interesting open problem. Social impact is out of the scope of the paper.

**Strengths And Weaknesses:**

Strengths:

1. Strong results, that compare favorably even to concurrent works.
2. Technically solid paper. Many interesting ideas which may be for independent interest.
3. Although the motivating question is more conceptual than practical, I still think it is worth of the attention of the machine learning community.

Weaknesses:

1. I find the idea of an ERM oracle quite unappealing. The optimization problems that ERM entail can be quite difficult to solve (specially for binary classification). So, although I believe that the reduction question is interesting on its own, it is hard to believe this model has any concrete consequences for ML.
2. Bounds depending on combinatorial parameters, such as VC dimension, can be quite pessimistic, and can be majorly improved by distribution dependent quantities, such as margin conditions. Maybe the results of the paper can easily transfer to these more benign characterizations, but it is not evident from the analysis.

---

> ### Author Response · Authors · 2022-08-02
> **Response to Reviewer4m7m**
>
> We thank the reviewer for the thoughtful review and the recognition of the importance of our work. We have added the references on transductive learning and made notational changes according to your comments in the revision of our paper. We discuss the other mentioned weaknesses below.
>
> **On use of Distribution-dependent Complexity Measures**: Essentially, all our results can be stated in terms of Rademacher complexity which is data dependent. In fact, our bounds depend on the Rademacher complexity of $T/\sigma$ i.i.d. instances with respect to the base measure, which is usually taken to the uniform measure on some structured set. We have clarified this in the revision.
>
> **On the motivation and applicability of the ERM oracle**: Please see our general comments for a detailed explanation.

---

> > ### Comment · Reviewer_4m7m · 2022-08-06
> > **Answer to authors' response**
> >
> > Thank you for your clarifying comments. I agree with them, and I am maintaining my score.

---

### Official Review · Reviewer_E2JM · 2022-07-12

**Rating:** 8
**Confidence:** 3
**Soundness:** 3 good
**Presentation:** 3 good
**Contribution:** 3 good

**Summary:**

For online learning with smoothed adversaries, optimal statistical
regret rates are known, but there are no efficient algorithms. Assuming
access to an offline ERM oracle, the present paper develops new
online algorithms that are computationally efficient (making 1 or 2
calls to the oracle per round, with the input size to the oracle also
under control). These algorithms achieve expected regret rates that are
optimal except for their dependence on the parameter sigma that measures
the smoothness of the adversary. There is also a lower bound that holds
for the expected regret for any algorithm with run-time below a certain
threshold.


**Questions:**

* Could you give examples of offline oracles that are sufficiently
  efficient to make your algorithms practically feasible?
* For comparison with the result from Theorem 5.1: what is the running
  time of Algorithms 1 and 2 in your model of "running time"?



**Limitations:**

* The efficiency of the algorithms relies on the availability of an
  efficient offline oracle, but the paper does not discuss for which
  hypothesis classes such offline oracles are available. In particular
  for the binary classification case, this might be an issue, since
  computing the ERM is NP-hard for many common classes.

* The algorithms require knowledge of sigma, which would typically not
  be available in practice.

**Strengths And Weaknesses:**

Strengths:
* Smoothed adversaries are seeing a lot of recent interest, but to move
  towards practically relevant results it is essential to have efficient
  algorithms, which is what the current paper tries to address.
* The algorithms are elegant reductions to calls to the oracle.
* Their analysis is clean, and clearly explained.

Weaknesses:
* I am not sure that the proposed algorithms will be so efficient that
  they can actually be used in practice, because implementing the oracle
  might still be a bottleneck.
* The lower bound in Thm 5.1 seems rather weak because the run-time
  requirement of omega(sqrt{d/sigma}) does not grow with the total
  number of rounds T.

Remarks:
* Line 125: mathcal{Y} cannot be an arbitrary subset of [-1,+1], because
  the predictions of Algorithm 1 can span the whole range [-1,+1].

Minor comments:
* Line 101: "unaviodable"
* Line 125: mathcal{Y} should not be an element of [-1,1], but a subset
* Line 128: For Y = {-1,+1}, convexity and Lipschitzness make no sense.
  (I assume these requirements do not apply in this case.)
* Paragraph below Definition 2.1: Equivalence between the two versions
  of the protocol is not obvious, because it seems to depend on whether the
  adversary and the learner are allowed to randomize, which is not
  specified explicitly.
* Multiple uses of "arginf" throughout the paper. Use "argmin" instead,
  because it only makes sense if the minimum is achieved.
* Line 179: Please specify distribution of \mathcal{E}^(t)
* Line 181: "labled"
* Multiple places: "\tilde{O} hideS factors"
* Theorem 3.2 and Corollary 3.3: regret -> expected regret.
* Section 4.2: historically, the perturbations for Follow the Perturbed
  Leader were not of the type described here, so perhaps clarify that
  your description is a special case.
* Line 282: please point out that beta is o(1/T). I initially thought it
  was of order T.
* Theorem 5.1: please explain how "total running time" is measured
  (e.g., copy explanation from the appendix)
* Line 522: I think you meant that beta = o(1/T) instead of o(T).
* Line 841, "both lower bounds": there is no lower bound on the regret
  in Theorem 5.1
* Lemma E.2: "supreme" -> "supremum"

---

> ### Author Response · Authors · 2022-08-02
> **Response to Reviewer E2JM**
>
> We thank the reviewer for the positive feedback and detailed comments. We have addressed the minor comments in the revision. Below we address specific comments and discuss the mentioned limitations.
>
> **On the efficiency and relevance of oracles to practical ML**: Please see our general comments for a detailed explanation.
>
> **About running times**: The notion of running in our Theorem 5.1 and 5.2 refers to the total number of operations performed by our algorithm, in which each oracle call takes unit time, and maintaining each element in the input to the oracle also takes unit time. In this model, the running time of Algorithm 1 is $\widetilde{O}(T^2/\sigma)$, and the expected running time of Algorithm 2 is $\widetilde{O}(T^2/\sqrt{\sigma})$.
>
> In Theorem 5.1, the reason that running-time requirement $\omega(\sqrt{d/\sigma})$ does not depend on $T$ is because we are more interested in the regime $d/\sigma \gg T^2$, where regret lower bound is sublinear. In this regime, the running times of our Algorithm 1 and 2 are $\widetilde{O}(d/\sigma^2)$ and $\widetilde{O}(d/\sigma^{3/2})$ respectively. We remark that Theorem 5.1 shows a poly($d, 1/\sigma$) computational lower bound to achieve the statistical regret $O(\sqrt{Td\log(1/\sigma)})$, while the $\epsilon$-net argument in [HRS21] requires an exponential poly($\sigma^{-d}$) computational time. From this observation, we pose an open question about whether $\Omega(poly(T, 2^d, 1/\sigma))$ computational time is unavoidable to achieve $o(T)$ regret in the smoothed analysis setting, even given access to an oracle.
>
> **Unknown the value of sigma**: Please see our general comments for a detailed explanation.

---

> > ### Comment · Reviewer_E2JM · 2022-08-08
> > **Answer to authors' response**
> >
> > Thanks for your response.
> >
> > * Your reply about adapting to sigma is fully convincing.
> > * I also agree with your view on the relevance of oracles to practical ML. It would be nice to include some of this discussion in the paper, which I believe would make it stronger.
> > * Thanks for clarifying the role of the lower bound.

---

### Author Response · Authors · 2022-08-02
**General Comments: Applicability and the Context of Oracle-efficient Online Learning**

**Applicability of oracle-efficiency to practical ML**: The oracle-efficient framework is practically important because it allows us to directly tap into existing deployed algorithms, without having to design and implement an algorithm from scratch. These sub-routine algorithms can be heuristics and do not have to be provably efficient.  Modern computer science is full of such heuristics that perform exceedingly well in practice even when NP hardness barriers exist in theory; a great example of this is the poster child for NP hardness, SAT. The oracle-efficient method for designing online algorithms has been extremely popular recently and has seen a lot of use in varied contexts such as contextual bandits and reinforcement learning (see https://vowpalwabbit.org/) and is even used in production. We see our work as putting forth a general framework on the instances under which we can design such online algorithms (for general learning problems).

**Sufficiency of weaker Oracles and circumventing Hardness**: Note that the oracle in our algorithm is called on either “smoothed” instances given by the adversary, or random instances sampled from the uniform distribution. In such settings, NP hardness results usually do not hold since they are proven mostly for worst case instances. Therefore, when implementing our algorithms in practice, instead of using an oracle that is provably efficient for all worst-case inputs, it suffices to have a weaker oracle that performs reasonably well on “average” case instances. In fact, NP hardness should not be a prime concern for assessing our results. Beyond the worst-case settings, the complexity landscape is in fact much richer — with hardness much harder to come by — and remains an active area of research (see  https://simons.berkeley.edu/programs/si2021 ). This is due to the fact that “average” case instances are much easier to handle, and can be seen as implicitly being the driver of the current machine learning revolution. This gives us hope that our methods can even be provably efficient for many hypothesis classes.

**History of Oracle-efficiency**: The oracle efficiency framework has a rich history in the online learning literature beginning with the work of Kalai & Vempala 2007, Awerbuch & Kleinberg 2007. The original motivation was problems such as online shortest paths, tree update and adaptive Huffman coding where the “set of experts” was combinatorially large but solving the corresponding offline problem was efficient. See Kalai-Vempala 07 for a great discussion on the original motivation. This framework has been built on and extended in various settings such as auctions [Dudik et. al, 2017] , contextual learning [Syrgkanis, Krishnamurthy & Schapire 2016, Foster & Rakhlin 2020, Simchi-Levi & Xu 2021], approximation algorithms [Kakade, Kalai & Ligett 2007], reinforcement learning [Foster, Kakade, Qian, Rakhlin 2021] etc. Note that, since the offline problem is at least as difficult as the online problem, in some sense an efficient oracle for the offline problem is the minimum requirement to hope to have an efficient online algorithm.

---

### Author Response · Authors · 2022-08-02
**General Comments: Knowledge of Sigma**

We clarify that the exact knowledge of $\sigma$ is not needed by our approach. Our algorithms and regret bounds can work with any approximation of sigma value that is a lower bound of the real $\sigma$ up to constant multiplicative factors. This corresponds to settings where the world is more smooth than we give it credit.
Even when we have extremely poor upper and lower bounds, we can use hedging (as suggested by reviews) to still get non-trivial regret with only a minor blow up in computation.  We will provide more details next as to how we work with knowledge of approximate sigma.

In general, given (loose) upper and lower bounds on the exact value of $\sigma$ (call them $\sigma_{\max}$ and $\sigma_{\min}$), we can use a geometric doubling approach to deal with the unknown $\sigma$. To be specific, one could construct $\log(\sigma_{\max}/\sigma_{\min})$ experts, where each expert runs a local version of our algorithm with parameter $\sigma_i=2^i \sigma_{\min}$. We then run Hedge on these experts. Note that the parameter of the best expert satisfies $\sigma/2 \le \sigma_{i*} \le \sigma$, so its regret matches the regret of the same algorithm that runs on true $\sigma$ up to a constant factor. Therefore, the expected regret of this meta algorithm is comparable to the bound in Theorem 3.1 and 3.2, with an additive term of order at most $\sqrt{T\log(\log(\sigma_{\max}/\sigma_{\min}))}$.
The number of oracle calls also blows up only by $ \log ( \sigma_{\max} / \sigma_{\min} )  $ per round. This could potentially be improved using a more aggressive step size for the Hedge meta algorithm.

In addition, it is natural to assume knowledge of $\sigma_{\max}$ and $\sigma_{\min}$. For instance, when smoothness comes from Gaussian perturbations, $\sigma$ directly relates to the standard deviation of the Gaussian distribution as well as the dimension of the instance domain. In this case, it is reasonable to assume that confidence intervals of the variance estimation are known in advance.

---

### Author Response · Authors · 2022-08-02
**General Comments**

We thank all the reviewers for their thoughtful feedback. In general comments below, we discuss two questions raised by multiple reviewers. The more specific reviewer comments are addressed separately below.

---

### Meta-Review · Area_Chair_mXFQ · 2022-08-24

**Recommendation:** Accept
**Confidence:** Certain

**Metareview:**

As summarized very well in the reviews, this is a well-written paper that makes a solid and elegant contribution to the recently active line of work on smoothed online learning.  The authors have successfully addressed the main concerns brought up in the discussion.  I genuinely agree the paper should be accepted.

As a side note to the authors: I honestly found your reaction to Reviewer gcMM’s comments rather aggressive and incongruous.  Disagreements naturally arise in a discussion and should not be automatically considered as an attempt to “greatly harm the review process and the community at large”.

**Award:**

No

---

### Decision · Program_Chairs · 2022-09-14

Accept